# Distinct functions for beta and alpha bursts in gating of human working memory

Johan Liljefors [1], Rita Almeida [1,2], Gustaf Rane[1], Johan N. Lundström [1,3], Pawel Herman [4] & Mikael Lundqvist [1] ✉

Multiple neural mechanisms underlying gating to working memory have been proposed with divergent results obtained in human and animal studies. Previous findings from non-human primates suggest prefrontal beta frequency bursts as a correlate of transient inhibition during selective encoding. Human studies instead suggest a similar role for sensory alpha power fluctuations. To cast light on these discrepancies we employed a sequential working memory task with distractors for human participants. In particular, we examined their whole-brain electrophysiological activity in both alpha and beta bands with the same single-trial burst analysis earlier performed on non-human primates. Our results reconcile earlier findings by demonstrating that both alpha and beta bursts in humans correlate with the filtering and control of memory items, but with region and task-specific differences between the two rhythms. Occipital beta burst patterns were selectively modulated during the transition from sensory processing to memory retention whereas prefrontal and parietal beta bursts tracked sequence order and were proactively upregulated prior to upcoming target encoding. Occipital alpha bursts instead increased during the actual presentation of unwanted sensory stimuli. Source reconstruction additionally suggested the involvement of striatal and thalamic alpha and beta. Thus, specific whole-brain burst patterns correlate with different aspects of working memory control.

Working memory (WM) is a key cognitive component that allows us to hold and manipulate information online in our mind[1–5]. WM has a very limited capacity and thus we need to control that only relevant information enters our WM[2,6,7]. Work on non-human primates and humans has, however, suggested distinct mechanisms of such WM control. Here we resolve this discrepancy and demonstrate multiple novel task- and region-specific neural correlates of human WM control on the single-trial level.

We previously observed high power beta (14–35 Hz) bursts in prefrontal cortex of non-human primates as a single-trial correlate of executive WM control[8–12]. At the conceptual level, these studies implied that cognition is supported by brief events rather than slowly changing dynamics and proposed that burst analysis may better capture moment-by-moment interactions between regions as cognitive commands unfold[13–15]. Specifically, beta bursting was reduced when information was encoded, particularly in cortical sites where neurons subsequently retained the WM information[8,12]. This suggests that spatial patterns of prefrontal beta bursts reflect the filtering of unwanted stimuli. Similarly, prefrontal beta bursts were suppressed when information in WM was accessed and subsequently elevated when it was cleared out.

In contrast, human neuroimaging studies focused on alpha (8–12 Hz) power in occipital and parietal areas as a correlate of filtering out visual information[16–21]. In particular, alpha power was shown to

[1]Department of Clinical Neuroscience, Karolinska Institutet, Stockholm, Sweden. [2]Stockholm University Brain Imaging Centre, Stockholm University, Stockholm, Sweden. [3]Monell Chemical Senses Center, Philadelphia, PA, United States of America. [4]School of Electrical Engineering and Computer Science, and Digital Futures, KTH Royal Institute of Technology, 10044 Stockholm, Sweden. ✉e-mail: Mikael.Lundqvist@ki.se

increase during the processing of irrelevant versus relevant stimuli. This elevation occurred in specific cortical locations depending on the visuo-spatial location of distractors or their sensory modality[19,20,22–24]. In addition, alpha oscillations in human occipital and parietal regions were considered to have an inhibitory role in other WM processes such as removal and selective prioritization of information[25,26].

Taken together, the control-related patterns observed in beta bursting in the frontal cortex of non-human primates[8,10,12] appear to be analogous to the ones observed in parieto-occipital alpha power fluctuations in humans[19,23,24]. This difference in frequency could be attributed to species differences, analysis methods (power fluctuations vs bursts) or the distinct areas studied. The observation that beta oscillations are gradually shifted towards slower frequencies lower in the cortical hierarchy, with the alpha dominance in visual area V4, would suggest the latter[9,27–29].

To test the hypothesis about regional specificity of WM control-related alpha and beta activity in human participants, we deployed a sequential WM task that required gating of input to WM. To determine single-trial neural correlates of WM gating, we recorded whole-scalp magnetoencephalogram (MEG) and frequency tagged each WM item in the sequence[21,30,31]. Frequency tagging involved modulating the luminance of stimuli at a known frequency entraining cerebral activity. The resulting frequency specific entrainment allowed us to estimate how strongly individual items were processed depending on their status as a target or distractor, and how this related to alpha and beta bursting in different cortical regions. The goal was to reconcile the proposed roles of beta bursting in macaques with alpha power in humans.

In particular, we asked whether both alpha and beta reflect the gating of distractor items? Do they correlate with gating information into WM observable on single trials and in behaviour? If so, do they have distinct roles and cortical origins? We found evidence that high-power bursts of both alpha and beta gate information in and out of WM but with partially distinct roles between the two frequency bands and between cortical locations. Overall, alpha seemed consistent with the suppression of sensory processing, whereas beta bursts more selectively appeared to gate information from sensory processing into WM and proactively removed information already retained in WM.

## Results

We recorded MEG data from 17 healthy volunteers while they performed a serial WM task (Fig. 1). The task required them to hold fixation on the centre of the screen throughout the whole trial. There were two sets of trials, with or without distractors. On No-Distractor trials,

four randomly oriented bars were sequentially presented foveally for 500 ms with inter-stimulus periods of 500 ms showing a fixation dot. Each bar was associated with a unique colour. Following a delay of 750 ms, the fixation dot turned into one of the four colours, acting as a retro-cue signalling which of the four bars would be tested. After an additional 750 ms, a test probe appeared on the screen and subjects were to rotate the test probe using a control pad to match the orientation of the memorized and cued bars within 5 s. On Distractor trials bar stimuli 2 and 3 acted as distractors, which should not be remembered and hence were never probed. Distractor and No-Distractor trials were randomly interleaved and were discriminated by a pre-cue just prior to each trial (Fig. 1).

### Behaviour suggested that distractors were successfully filtered out

To establish that subjects correctly performed the task and selectively encoded target bars, we first analysed behaviour. We measured performance as the absolute value of the angle between the orientations of the probed and reported bars. The data was modelled using a linear mixed-effects model, with subject as random effect ($n = 17$). To meet the general linear model assumptions the performance data was first transformed (see Methods). Condition (Distractor or No-Distractor), order of the bar probed and the interaction between these two variables were modelled as fixed effects. In the first model, trials probing bar 2 and 3 in the No-Distractor condition were removed to match the trials of the Distractor condition. Effect of condition was significant (beta = 0.56539, t(5080) = 9.89, $p < 2e\text{-}16$), as well as order (beta = -0.46529, t(5080) = -9.968, $p < 2e\text{-}16$). Further, the interaction between condition and order of the probed bar was significant (beta = -0.42525, t(5080) = -5.26, $p < 2e\text{-}7$), meaning that the effect of condition was not the same for bar 1 and bar 4. The performance in the Distractor condition was better than on the No-Distractor condition, although the task difference for bar 1 (beta = 0.5654, t(2532) = 9.585, $p < 2e\text{-}16$) was larger than for bar 4 (beta = 0.1401, t(2352) = 2.554, $p = 0.011$). This suggested that participants indeed treated bar 2 and bar 3 as distractors on these trials, freeing up resources to encode the two target bars on Distractor trials with higher precision.

Separating the data according to the condition, we found a clear serial order effect for both conditions, with much smaller errors on the last bar in the sequence relative to the first bar (Distractor trials (beta = -0.46529, t(3382) = -10.2, $p < 2e\text{-}16$), No-Distractor trials (beta = -0.89054, t(1682) = -12.96, $p < 2e\text{-}16$). Finally, we analysed performance on the No-Distractor condition in relation to the bar probed, considering all four bars. There was a significant effect of bar ($p < 0.0001$).

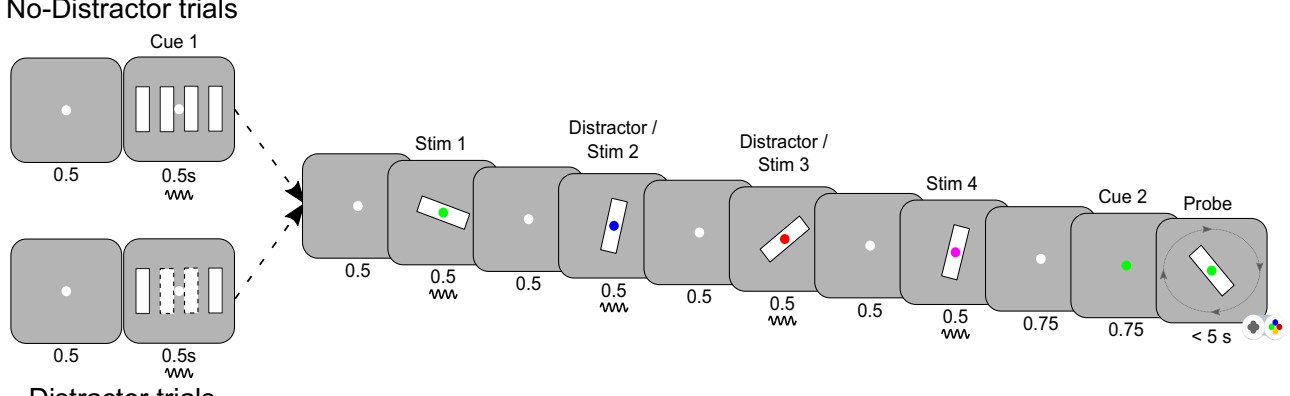

**Fig. 1 | Experimental design.** A sequential array of four bars (stim 1-4) with random orientation was presented. Each bar was randomly associated with a colour (red, green, blue or magenta dot in the middle of the bar). Later, a retro-cue with one of the four colours was presented, identifying the bar orientation to recall. In half of

the trials (randomized), bar 2 and 3 in the sequence were never tested and acted as distractors. These trials were indicated by a pre-cue. The pre-cue and all four bar stimuli were frequency tagged (Methods). The subjects had 5 s to submit a response before the next trial. Each subject performed 400 trials.

**Fig. 2 | Performance by trial type and the sequential position of the bar stimulus.** Average absolute errors (angular distance between probed angle and response) are shown with their ±1 standard error of mean (SEM, $n = 17$ subjects), for each bar in the sequence and by trial type (left Distractor trials, right No-Distractor trials). Significance analysis was conducted using linear mixed-effects models, which included fixed effects for trial type and sequential position, and random effect for subjects. Planned comparisons were conducted using Tukey's method. All hypothesis tests were two-sided. Significance levels are indicated as follows: *** denotes $p < 0.001$, * $p < 0.05$. Other statistical details are provided in the main text. Source data are provided as a Source Data file.

Planned comparisons using Tukey's method showed that the effect was driven by better performance on the last bar when compared to the three first ones, for which there were no statistically significant differences in performance (Fig. 2).

Taken together, the results suggested that the subjects appeared to encode all target bars and selectively skipped to encode distractors. The encoding of additional bars in the sequence in No-Distractor trials degraded the memory representations of the bars already encoded, leading to a strong order effect. Thus, for the last bar in the sequence there was relatively good performance regardless of trial type.

**Frequency tagging revealed correlates of selective encoding**
We next analysed the MEG signals at the sensor level. Each oriented bar stimulus and the pre-cue were frequency tagged (alternating between 31.1 or 37.1 Hz, see Methods). Because the frequency tagging was phase-locked to the stimulus onset, we separately analysed total, phase-locked and induced power. Neural activity corresponding to the processing of the tagged stimuli would be most strongly observed in the phase-locked power as the tagging itself was phase-locked to the stimulus onset (Fig. 3a; see contours $p < 0.05$). We assessed neural substrates of gating by contrasting power in the No-Distractor and Distractor trials. For phase-locked power this analysis demonstrated that distractors entrained cortical activity in the tagged frequencies to a significantly lower degree than targets around stimulus presentations (Fig. 3a). The topography of these effects suggested that the main difference in entrainment of tagged frequencies were around the occipital sensors (Fig. 3b). In the induced power, more strongly reflecting endogenous oscillations, we observed significant differences in the alpha and beta frequency ranges between these two conditions around the same time (Fig. 3a). Thus, it suggested the involvement of alpha and beta oscillations as correlates of filtering information into WM.

**Alpha and beta bursts correlate with selective suppression of distractors**
To directly connect our current results to prior work on beta bursts in non-human primates[8,12], we extracted bursts in three frequency ranges (alpha: 8-12 Hz; low beta: 12-18 Hz; high beta: 18-26 Hz) for each sensor (Methods). The choice of distinct frequency bands were motivated by our recent findings in non-human primates that the "functional" beta frequency tend to be lower further down in the cortical hierarchy[9]. Thus, we suspected that there could be important differences between sub-bands in terms of cortical origin and function. Further, our analysis was bounded upwards to avoid overlap with the tagged frequencies. The average burst rates, for all sensors and all trials, for both alpha and beta frequencies showed strong similarities with prefrontal beta burst rates in non-human primates (Fig. 4a, b). Namely, the burst rates were elevated during fixation and delays, and suppressed during stimulus presentations. This is generally consistent with their proposed inhibitory role in filtering cortical bottom-up processing[8,12]. There were however two striking differences between the alpha and beta burst rates over time. First, alpha peaked just before stimulus onset, was suppressed during each stimulus presentation but then started to smoothly rebound and peak just before the next presentation. This rebound started even before the stimulus was removed. In the beta bands, the pattern was similar but whereas alpha had preserved levels of bursting between the different stimuli in the sequence, beta burst rates were gradually lower as the sequence progressed (Fig. 4b, c). One-sample t-tests confirmed a negative burst gradient for low beta ($t(16) = -4.03$, $p < 0.001$, Cohen's $d = 0.93$) and high beta ($t(16) = -7.04$, $p < 0.001$, Cohen's $d = 1.63$), while alpha was non-significant

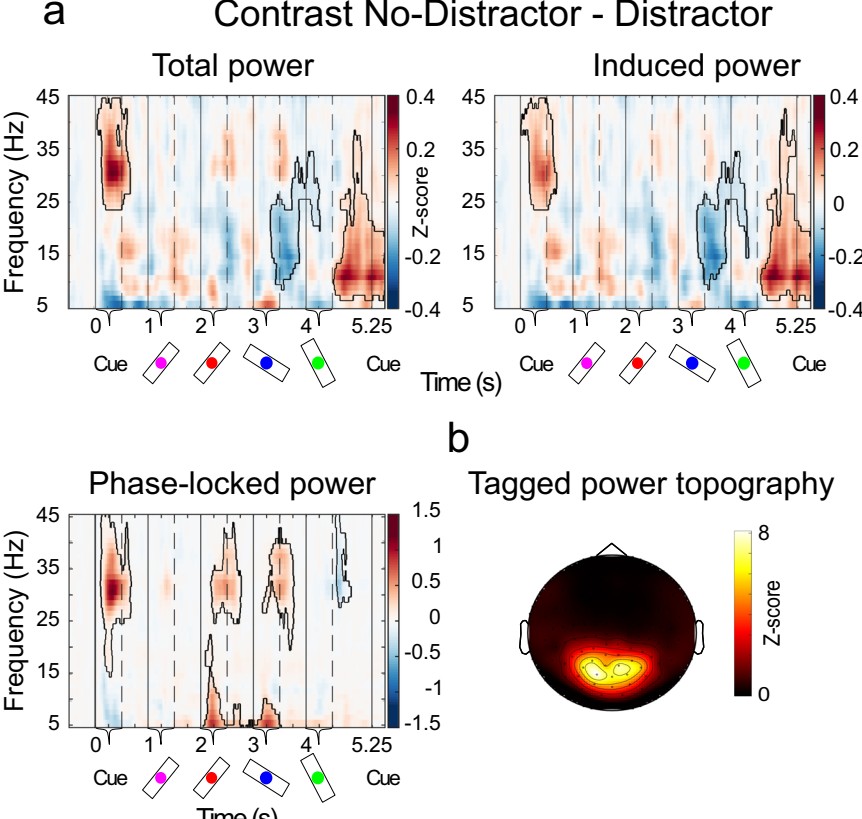

**Fig. 3 | Time-frequency analysis of Distractor and No-Distractor trials. a** Power contrasts between the Distractor and No-Distractor trials. Spectral decompositions were done using adaptive superlets and z-scores were calculated on a trial level normalized to a pre-stimulus window of 250 ms before the onset of the cue (see Methods). Induced power was calculated by subtracting the event-related field from the raw MEG data before spectral decomposition and phase-locked power as the difference between total and induced power. Vertical lines show the onset/offset of the cues and bars. Boundaries delineate areas as statistically significant using cluster-based non-parametric tests at the 95% confidence level. **b** The topographical sensor plot shows the average phase-locked power for the tagged frequencies (31.1 Hz and 37.1 Hz).

($t(16) = -1.39$, $p = 0.09$, Cohen's d = 0.32). We have previously observed the same for beta bursting in the prefrontal cortex of non-human primates[8]. Second, in contrast to alpha, beta burst rates were timed to both stimulus onset and offset, not just onset (Fig. 4d). Both these differences were clearer in the higher beta band compared to the lower one, gradually changing from alpha to higher beta.

### Alpha and beta bursts exhibited distinct functions depending on cortical origin

To better understand the relevance of burst patterns observed in the alpha and beta bands for WM filtering, we next investigated how they differed in occipital, parietal and prefrontal sensors, and between Distractor and No-Distractor trials (Fig. 5; see Figure S1 for the same analysis based on power). There were several significant differences between the two conditions. First, there was increased beta and alpha bursting during and immediately following the presentations of distractors when compared to targets (Figs. 5, 6). This difference was only seen in occipital sensors (Fig. 6a; main significant clusters: alpha $p$ value < 0.001, low beta $p$ value < 0.001, high beta $p$ value < 0.001; see Figure S2 for parietal and prefrontal sensors) and coincided with the decreased processing of distractors as measured by the phase-locked power in the tagged frequencies over occipital sensors (Fig. 3a). The elevated bursting in occipital sensors during distractors had different characteristics depending on the frequency band (Fig. 6a). Alpha bursting was more suppressed during and following the presentation of targets relative distractors. Beta bursting, by contrast, had a distinct peak at around the time when distractor

presentations ended (Figs. 5, 6a). This was consistent with their distinct temporal dynamics, i.e. beta (high beta in particular) was timed to both the onset and the offset of stimuli and not just the onset as in the case of alpha. Thus, the difference in beta bursting between distractors and targets was more focused around the offset of the distractor, in the transition from sensory to WM processing.

Second, in contrast to the above, there was less bursting in Distractor trials in the delay periods preceding the distractors compared to the delay period prior to the corresponding target stimuli (bar 2 and 3) in No-Distractor trials. This was primarily seen in parietal and prefrontal sensors and mostly in the beta bands (Fig. 6b; parietal main significant clusters: low beta $p$ value < 1e-6, high beta $p$ value < 1e-6, PFC main significant clusters: low beta $p$ value = 0.001, high beta $p$ value < 1e-6; similar qualitative differences in alpha were not significant, see Fig. 5a and Figure S2). Thus, in the period before target items were about to be presented, there was elevated bursting in prefrontal and parietal sensors. This is again consistent with the suggested inhibitory role of beta bursting: it proactively freed up resources in higher-order regions for the upcoming WM item by downregulating information already held in WM. Prefrontal sensors had overall similar behaviour as the parietal, with a key difference that there was also a brief period of elevated bursting after the second distractor consistent with disregarding or removal of that task-irrelevant information.

Third, the gradual decrease in beta bursting over the course of the sequence was only observed over parietal and prefrontal, not occipital sensors (Fig. 5b). One-sample t-tests showed no significant gradients in

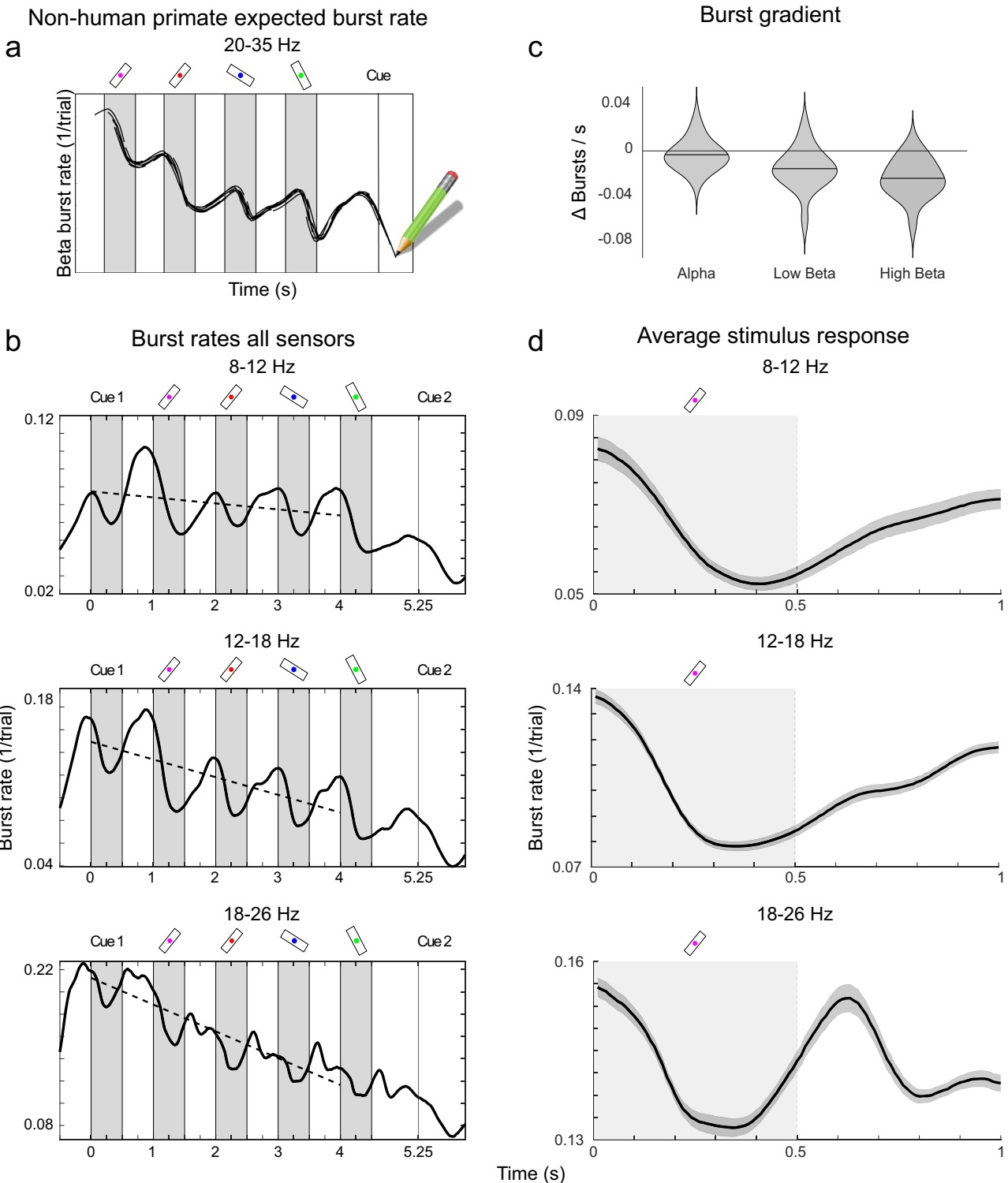

**Fig. 4 | Burst rates across all sensors. a** Expected beta burst rate in Macaque monkeys based on prior research where monkeys were tasked to memorize 3 stimuli for periods of 0.3 s[8]. The illustration has been adapted to be comparable to our experiment. **b** Distinct temporal dynamics of alpha and beta bursts across all sensors. Plots show the grand average burst rates (across all sensors, trial types and subjects) for the three frequency bands. Burst rate is calculated as the fraction of trials where a burst was detected at that given part of the trial (Methods). The greyed areas represent the bar stimulus presentation, while the dotted line indicates the linear trend for burst rate decay calculated using least squares. **c** Burst gradient by frequency band for all trial types, represented as the slope of the dotted lines in Fig. 4b. Violin plots are based on subject level averages, horizontal line indicates mean. **d** Burst rate from **c**, averaged over 1 s periods including the presentation and the following delay period for all 4 bar stimuli. Grey regions indicate stimulus presentation periods, error bands represent ±1 SEM. Source data are provided as a Source Data file.

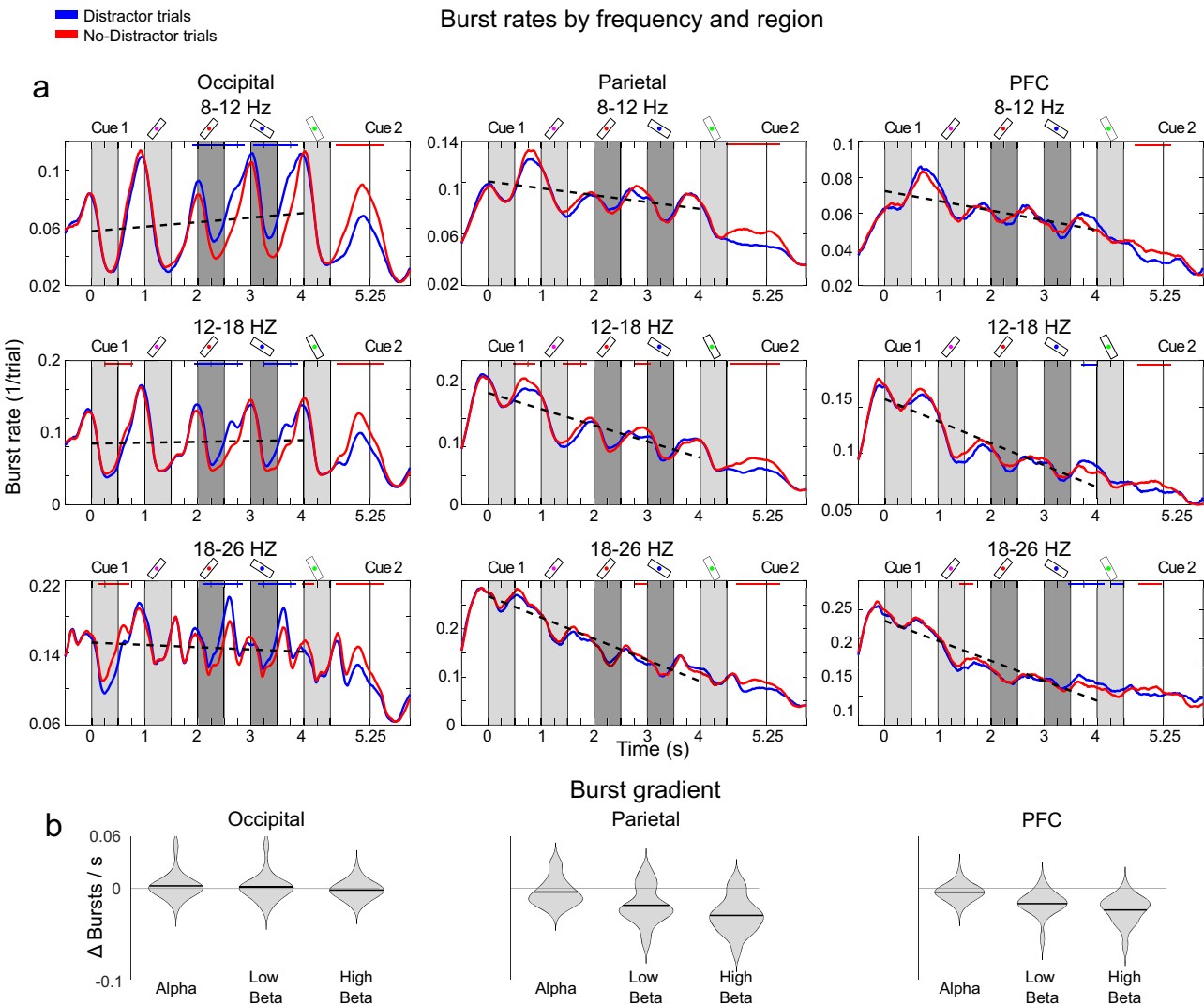

**Fig. 5 | Bursting by frequency and cortical region. a** Burst rates for trial types and regions. Burst rates per frequency band (rows) and region (columns) are shown. Burst rates for Distractor (blue) and No-Distractor (red) trials are plotted independently. Blue bars denote when Distractor trial burst rates were significantly above No-Distractor trial burst rates, and red bars the opposite, using two-sided cluster-based permutation test at the $p < 0.001$ level. The dotted line indicates the burst gradient, calculated through a linear fit using least squares to all trials. Distractor presentation periods in Distractor trials (bar 2 and bar 3) are indicated by dark (as opposed to light) grey areas. **b** Burst gradient by frequency band and region for all trial types, represented as the slope of the dotted lines in a. Violin plots are based on subject level averages, horizontal line indicates mean. Source data are provided as a Source Data file.

occipital cortex: alpha (t(16) = 0.71, $p = 0.76$, Cohen's d = 0.16), low beta (t(16) = 0.31, $p = 0.62$, Cohen's d = 0.07), high beta (t(16) = -1.21, $p = 0.12$, Cohen's d = 0.28). In parietal cortex there were negative gradients for low beta (t(16) = -4.22, $p < 0.001$, Cohen's d = 0.98) and high beta (t(16) = -7.36, $p < 0.001$, Cohen's d = 1.70), but not for alpha (t(16) = -1.40, $p = 0.09$). In PFC there was a significant gradient in all frequency bands: alpha (t(16) = -2.34, $p < 0.05$, Cohen's d = 0.54), low beta (t(16) = -5.81, $p < 0.001$, Cohen's d = 1.34) and high beta (t(16) = -7.55, $p < 0.001$, Cohen's d = 1.74). Finally, in the delay period following the sequence, just prior to the retro-cue there was a larger elevation of bursts for No-Distractor trials in all three frequency bands and all sensors (see Fig. 5a for statistical differences at $p < 0.001$). This difference likely reflected WM load (2 vs 4 items), and disappeared after the retro-cue, when in both conditions only a single item remained relevant for the task.

### Source reconstruction suggested the involvement of sub-cortical structures

We performed the main analysis on sensor space data since it yielded robust differences between regions while requiring minimal

processing and choices in the analytical pipeline. However, source reconstruction may further highlight important differences between alpha and beta bursts. Using subject T1 scans, we applied minimum norm estimate (MNE) to project sensor level activity onto an MNI normalized source space with 1 cm³ resolution (Methods). The MNI source space was parcellated into volumes of interest using the AAL atlas, and for each volume we performed burst analysis. First, we compared our source reconstructed data to our sensor space findings by taking the average burst rates per condition across all volumes within our three regions of interest. This yielded overall highly similar results to the analysis performed in sensor space (Figure S3). However, a striking difference was in the occipital regions, where high beta bursts rates almost exclusively seemed to respond to stimulus offset, accentuating the differences with alpha burst rates in the same region.

Next, focusing on the differential bursting between distractor and target presentations we tested for significant clusters in each brain volume (from the AAL atlas) for each frequency band independently (using $p < 0.01$ for cluster threshold and significance alpha during

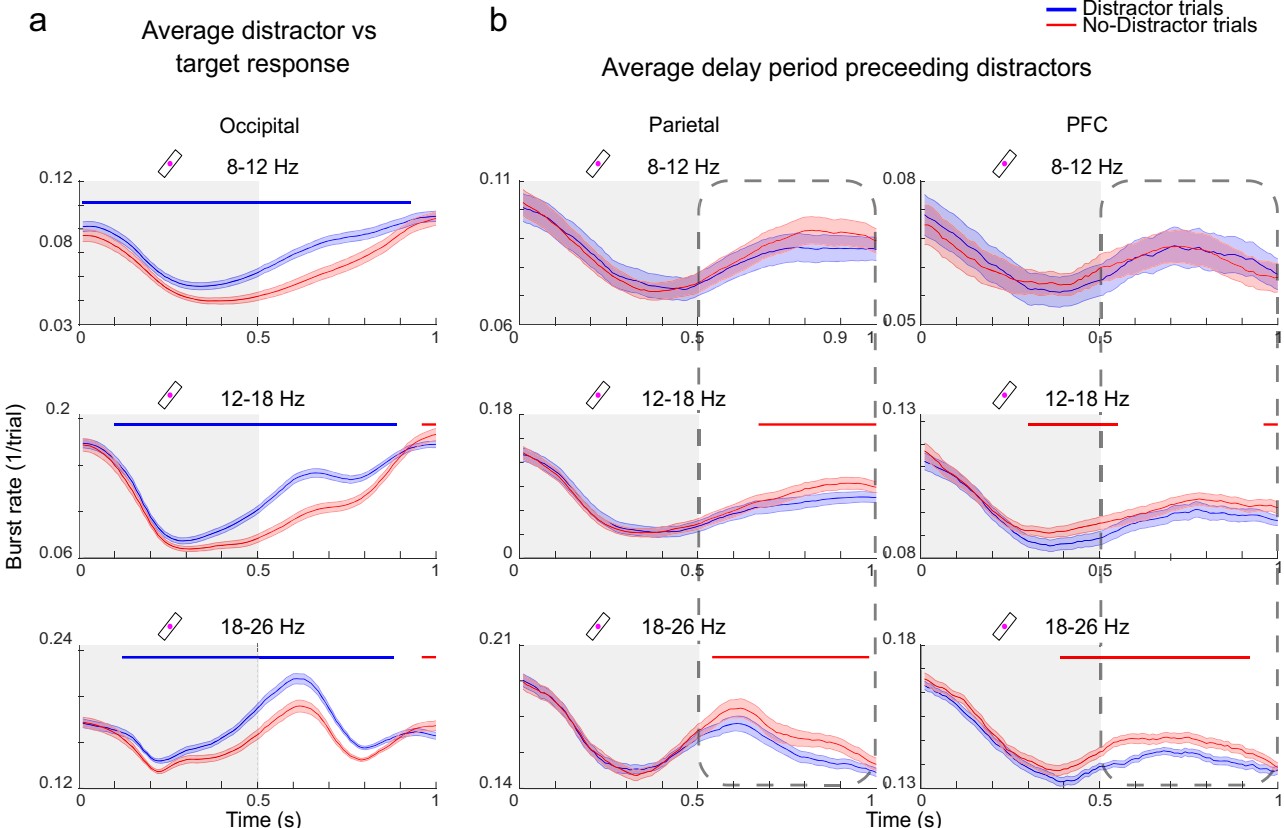

**Fig. 6 | Bursting before and after distractor presentations. a** Burst rates as shown in Fig. 5a but averaged over 1 s periods starting from the onset of bars 2 and 3 (serving as distractors in Distractor trials) for each frequency band and the occipital area. The grey shaded area indicates the presentation period of bars 2 and 3. **b** Burst rates as shown in Fig. 5a but averaged over 1 s periods starting from the onset of bars 1 and 2 for each frequency band and the parietal and PFC areas. The dotted boxes represent the delay periods preceding the presentation of distractors (blue, in Distractor trials) or targets (red, in No-Distractor trials) for bar 2 and 3 in the sequence. Blue bars denote when Distractor trial burst rates were significantly above No-Distractor trial burst rates, and red bars the opposite, using two-sided cluster based permutation test at the $p < 0.001$ level. Error bands reflect ±1 SEM. Source data are provided as a Source Data file.

encoding, see Methods). This revealed that effects were focused on the occipital volumes, particularly for alpha (Fig. 7a). For all three frequencies we also observed effects in thalamus and fusiform gyrus within temporal cortex (Fig. 7a). For beta frequency bursting we additionally observed differences in parietal and dorso-striatal volumes (Fig. 7a, b), with elevated bursting for distractors. The involvement of thalamus and striatum is of particular interest as we have speculated that beta bursts might arise from cortico-subcortical interactions[5,15]. To investigate temporal relationships between bursts in different brain regions we performed burst-triggered analysis between pairs of brain regions. It allowed us to discern if the onset of a burst in a certain region tended to precede or follow bursts in another. Overall, our findings suggested that bursts in prefrontal and occipital regions led parietal and sub-cortical bursts (Figure S4). These patterns were mostly stable over different phases of the task. However, prefrontal cortex was the highest in the hierarchy during encoding, and occipital cortex during retention.

**Beta and alpha bursts correlated with encoding on single trials**
The burst analysis offered an opportunity to study beta and alpha bursts as single-trial correlates of sensory filtering. We first analysed how the amplitude of phase-locked power in the tagged frequencies during stimulus presentations was modulated by the presence of bursts (burst-triggered phase-locked power). For this we used all bursts from occipital sensors, i.e. the sensors in which tagged frequencies were modulated by distractors, occurring during stimulus processing (between 200 to 700 ms from stimulus onset, the period in

which tagged frequencies were elevated). Since both burst rates and phase-locked power showed strong modulation over the course of a stimulus presentation, we compared burst-triggered power to surrogate data to reveal a single-trial relationship (Methods). The surrogate data was produced by shuffling trial indices (for each sensor, subject and condition independently) for which the associated phase-locked power was taken from. Thus, importantly, the temporal structure was preserved in surrogate data. This revealed that bursts in all three frequency bands had a small but significant inhibitory effect on phase-locked power (see Fig. 8 for significant clusters for 31.1 Hz tagging frequency; see Figure S5 for 37.1 Hz tagging frequency). Again, there were some differences between bands. For alpha, power was reduced following the onset of bursts. For beta, there was in addition significantly elevated phase-locked power prior to the onset of bursts. We interpreted this as beta bursts timely responding to the increases of power in tagged frequencies occurring around stimulus presentations, consistent with beta's role in active regulation of processing. As a control analysis we performed the same analysis but based on the frequency that was not tagged at each presentation (Figure S6). It revealed similar effects (likely resulting from spectral leakage from the tagged frequency) but an order of magnitude smaller, thus ruling out a possibility that the effects were based on spectral leakage from bursts themselves.

Finally, to relate bursts to behaviour in single trials, we constructed a linear mixed-effects model based on trials where probed items were the first or last bar. The performance (absolute magnitude of error) data was transformed to meet the general linear model

**Fig. 7 | Burst rates in source space starting from onset of bars 2 and 3 (serving as distractors in distractor trials). a** AAL regions with significant burst rate differences between trial types ($p < 0.01$) during bar 2 and 3 presentation periods. **b** Burst rates averaged over 1 s periods starting from the onset of bar 2 and 3. The grey shaded area indicates the presentation period of bars 2 and 3. Blue bars denote when Distractor trial burst rates were significantly above No-Distractor trial burst rates. AAL regions for Caudate (Caudate_L, Caudate_R), Thalamus (Thalamus_L, Thalamus_R) and Putamen (Putamen_L, Putamen_R). Test were done using two-sided cluster-based permutation test at the $p < 0.01$ level. Source data are provided as a Source Data file.

assumptions (see Methods). The model had subject as random effect, condition, serial order of probed item, and occipital burst count during encoding (onset to 500 ms) as fixed effects. We performed this analysis for each frequency band separately. It suggested significant relationships in all frequency bands (alpha burst count (coefficient = 0.844, t = 4.18, $p$ = 3e-5, df = 4908), probed item order (coefficient = -0.435, t = -15.87, $p < 2e-16$, df = 4896), trial type (coefficient = 0.343, t = 8.35, $p < 2e-16$, df=4896); low beta burst count (coefficient = 0.667, t = 3.42, $p$ = 0.0006, df = 4907), probed item order (coefficient = -0.430, t = -15.67, $p < 2e-16$, df = 4896), trial type (coefficient = 0.344, t = 8.37, $p < 2e-16$, df = 4896); high beta burst count (coefficient = 0.587, t = 2.71, $p$ = 0.0068, df = 4904), probed item order (coefficient = -0.428, t = -15.06, $p < 2e-16$, df=4896), trial type (coefficient = 0.345, t = 8.39, $p < 2e-16$, df=4896)), such that higher burst counts during encoding subsequently led to lower WM accuracy. In addition, we performed the same analysis but using burst counts in prefrontal and parietal regions preceding the second and third stimuli. It suggested weak evidence in the high beta band, which was not significant (alpha burst count: coefficient = 0.0030, t = 1.16, $p$ = 0.25, df=4910; low

beta burst count: coefficient = 0.148, t = 1.37, $p$ = 0.17, df=4906; high beta burst count: coefficient = 0.247, t = 1.89, $p$ = 0.06, df=4901).

## Discussion

We set out to resolve the distinct neural correlates of WM control proposed by earlier studies. Prior work has either suggested prefrontal beta bursts (non-human primates;[5,8,10–12]) or occipital power fluctuations in alpha (humans;[17,19–21,23,25,26,32,33]) as correlates of visual attention and WM control. We applied the single-trial burst analysis we had originally developed for the non-human primate data[8] on human data recorded with MEG. Overall, the quantification of cortical rhythmic activity with narrow-band bursts rather than with more traditional power has several advantages[15]. First and foremost, it is conceptually important whether discrete or slowly changing neural dynamics support cognitive processes. In addition, burst analysis provides enhanced signal-to-noise ratio by reducing the effect of aperiodic activity in the given band between the burst events. It also offers meaningful information about the timing of neural activity thereby helping to relate alpha/beta activity to behaviour or other metrics on a

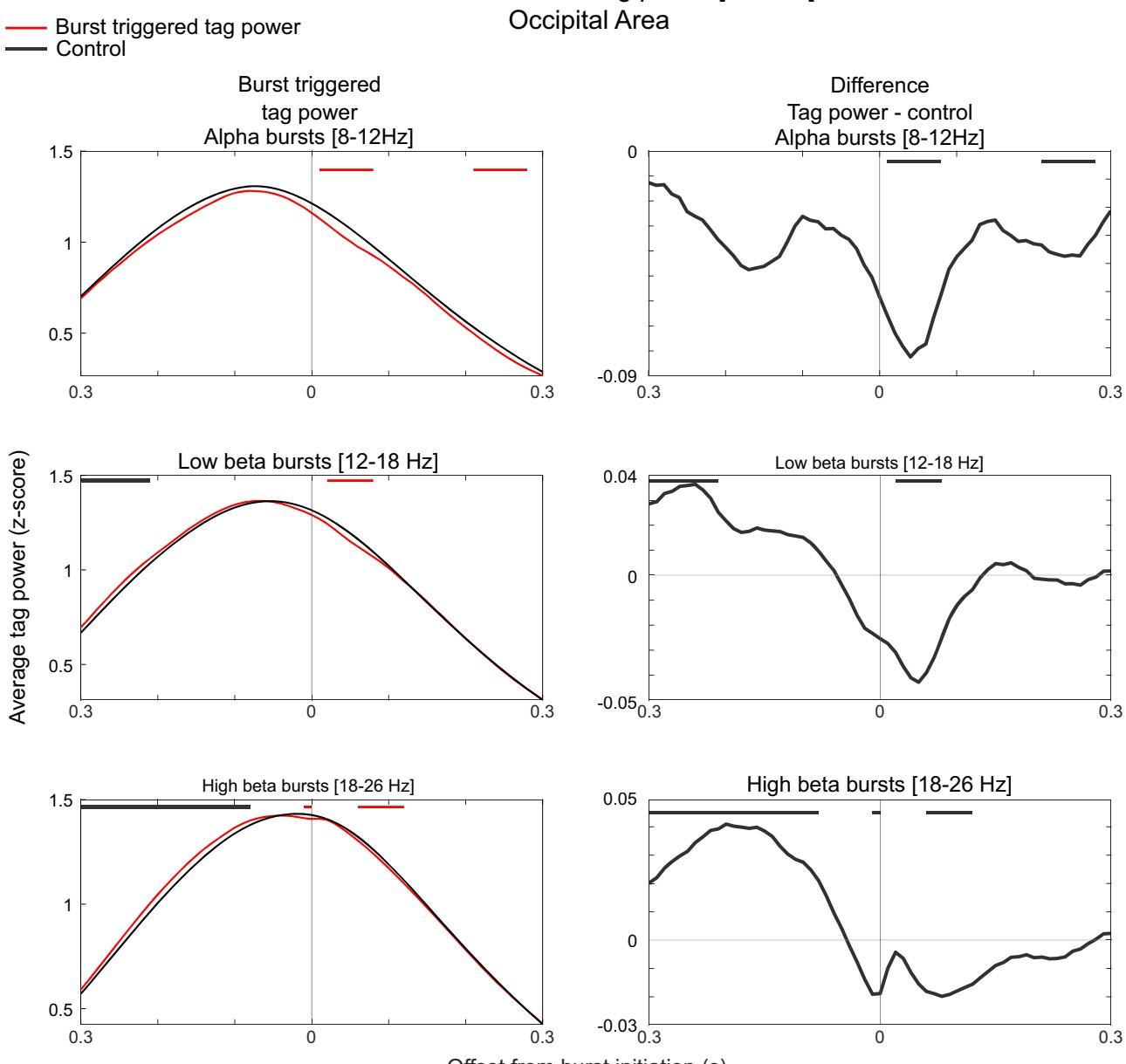

**Fig. 8 | Burst modulation of tag power for each frequency band.** Left column shows the burst triggered phase-locked power (31.1 ± 2 Hz, z-scored, see Methods) in red. Time 0 denotes the time of the detected burst onset. Displayed is also surrogate data, calculated by shuffling trial labels (within each condition and sensor independently) to estimate the power in the tagged frequencies drawn from the same distribution of times as the observed bursts (grey). Red bars denote periods in which power in the original data is lower than the surrogate data and grey bars indicate when power is higher in the original data than the surrogate data using two-sided permutation test at $p < 0.001$. The right column zooms in the difference between modulated tag power and surrogate data. The right column shows the difference between the actual data and the surrogate data. Source data are provided as a Source Data file.

moment-by-moment basis, more directly than smoother estimates of spectral power modulations.

Our single-trial burst analysis of MEG demonstrated that WM-related beta bursts had similar patterns in humans as in non-human primates. In addition, alpha bursts shared a similar behaviour to beta bursts but with some key distinctions. In particular, the reported contrasts between target and distractor processing were observed at different times and in different cortical regions for the two bands. Alpha bursts, primarily in occipital regions, suppressed distractor stimulus processing. Occipital beta bursts were instead elevated just as distractor stimuli were removed from the screen in the transition from sensory processing to WM retention. In addition, beta bursts in the

parietal and prefrontal regions tracked sequence order and were elevated *before* target items. The latter pattern is consistent with inhibitory beta bursts in higher-order cortex freeing up space in WM prior to additional encoding. Earlier studies have indeed suggested that while alpha appears to be important in the filtering of irrelevant sensory information there may also be several complementary mechanisms beyond gating at play[34–36].

Our existing non-human primate model suggests that the level of beta bursting reflects the level of inhibitory cognitive control[5,8,10,12]. This model is based on simultaneous analysis of spiking and intracranial LFPs on the single trial level, where beta bursts suppress gamma bursts and spiking. It also accounts for overall patterns of beta bursting

during various cognitive operations, and how they differ on recording sites in which information is or is not encoded into the patterns of spiking. Thus, in this model beta bursting is suppressed during encoding of information into WM, especially on recording sites where information is encoded, and during read-out of information from WM. Beta bursting has intermediate levels when information is retained but not used, and then strongly elevated following each trial when information has to be cleared out, in particular on those sites where the information was encoded[5,8,12]. Here we observed analogous beta and alpha burst rate patterns in occipital, parietal and prefrontal regions. While we reported task-specific modulation of alpha and beta bursting in most of the brain, the patterns were relatively weak in prefrontal regions, at odds with the strong focus on the prefrontal cortex in non-human primate WM[3,5]. This likely stems from the nature of MEG recordings with relatively low signal-to-noise ratio over frontal regions. However, the timing of alpha and beta bursts in prefrontal regions tended to precede bursts in other areas, including subcortical structures, suggesting a key role for prefrontal bursting. Additionally, a recent human study suggested that persistent training over time, as is typically the case in non-human primates, makes prefrontal WM representations more pronounced, providing a potential explanation for the differential focus on brain areas between species[37].

To establish a single-trial correlate of filtering of sensory information using only non-invasive measures we used frequency tagging[21,30,31] and also correlated bursts with behaviour. The latter revealed a single-trial link between the amount of bursting during presentation of target stimuli and subsequent precision in the WM task. Unlike in earlier studies, which analysed power fluctuations, we observed that occipital beta and alpha bursting suppressed sensory processing as estimated by tag power on the single-trial level[31,38]. Thus, our findings suggest a direct link between occipital alpha/beta bursts and cortical excitability. This is important as the lack of such a relationship calls for alternative interpretations of alpha and beta[36]. Here the burst analysis (as opposed to power analysis) bears particular relevance as there was only a transient window of reduced power at tagged frequencies following the onset of a burst. Thus, on the whole-trial level such relationship may be masked if, for instance, bursting is upregulated on trials in which there happens to be a strong cortical response to the stimulus. There was an interesting difference between beta and alpha bursts in this regard. Processing tended to be elevated just prior to each beta burst, and not just suppressed during their presence. We interpret this as beta being upregulated by a feedback mechanism when cortical levels of activity are too high. Alternatively, it was recently observed that beta bursts are preceded by a brief window of excitation before the longer-lasting inhibition[39]. This could also explain the elevated levels of processing preceding beta bursts.

There were several other differences between alpha and beta bursting, and between the cortical regions. We argue that these differences, taken together, suggest that occipital alpha reflects sensory filtering, while beta, particularly in higher-order areas, reflects instances when information is already being processed and cortical activity has to be subdued: occipital alpha was suppressed more during the presentation of target items compared to distractors. However, the overall rates were not modulated by other task factors such as load or the passage of time within trials. Occipital beta bursts rates were also modulated by the presence of distractors but more focused around the time of the removal of the distractor from the screen (thus when it would have to be encoded into WM). Activity over parietal and prefrontal sensors had quite a different pattern with significantly elevated beta bursting *before* the presentations of target items (as opposed to upcoming distractors). We interpret this as downregulation of existing WM-related activity to reallocate resources for the upcoming target items in a proactive way. We draw these conclusions primarily based on timing of the alpha and beta bursting. Further experiments, where the order of distractors and target items are varied, and in which

distractors are not always predictable, may shed further light on this potential distinction between alpha and beta bursting. If, for instance, distractors can appear both before and after target items are presented the proposed role of beta in regulating already encoded information may be directly tested.

In addition, beta (but not alpha) burst rates in parietal and prefrontal regions were strongly modulated by the passage of time within trials, with gradually lower burst rates as the trial advanced. Since this trend was equally strong in Distractor and No-Distractor trials we found it likely to be tracking the task structure over time, rather than being load dependent (which also changed with time but differently for the two types of trials). Thus, beta bursting in higher-order areas may be more directly linked to executive control functions, where keeping track of the various parts of a trial is essential. We have recently proposed that the spatio-temporal evolution of beta bursting patterns help implement cognitive operations by directing information flow to distinct patches of cortex during distinct parts of a task[10,15]. This shapes low-dimensional and task-related aspects of single neuronal spiking that are needed to solve the cognitive task at hand[40,41].

We also observed a somewhat puzzling finding from the perspective of beta and alpha bursting providing transient inhibition. While at the end of the delay, the retro-cue induced more alpha and beta bursting in no-distractor trials, consistent with more inhibition when more items need to be suppressed around the retro-cue, or protected from sensory interference, there were no load effects prior to the final delay (i.e. between Distractor and No-Distractor trials). It could potentially be that alpha and beta bursting during the encoding sequence mainly reflected executive control for allocation of the WM resources, which in total were similar in the two sets of trials (with more resources per item and better performance in load 2 trials). The consistent reduction in beta bursting observed in the parietal and prefrontal sensors (and sources) with the presentation of each item regardless of cognitive status as distractor or target, might reflect a mechanism of tracking the sequence of events during the trial, as discussed above. Prior studies have reported load dependent alpha and beta power during retention but without any analysis relating to the gradual build-up of WM load during the encoding phase[42,43].

The role of beta in WM executive function was also recently supported in clinical studies[44,45]. In patients with Parkinson's disease, insufficient cortico-striatal beta power suppression during encoding of information into WM was linked to diminished WM performance and correlated with symptom severity[44]. In obsessive-compulsive disorder, the diminished prefrontal beta power rebound following trials was linked to the impairment in the removal of information from WM[45].

Changes in alpha and beta power during stimulus processing have been extensively studied[46–49]. These dynamics have been attributed among others to encoding of information into episodic and long-term memory[50,51], and several theories have suggested a link between alpha and the dynamic functional architecture of cortex[47,48,52]. Specifically related to our findings, numerous human studies have suggested that modulation of alpha power reflects filtering of sensory processing in attention and WM tasks[16–18,20,23,24,31,49], although not necessarily under strict top-down control[21,36]. Intracranial recordings have demonstrated that alpha activity was finely tuned spatially, consistent with a role in the selective suppression of unwanted information in a visual scene[19,22]. Furthermore, alpha power was reported to reflect the modality-specific suppression of distractors in a WM task involving visual and auditory inputs[23]. A load-dependent upregulation of alpha during WM retention has also been observed, consistent with a role in protecting WM information from distractors[43,53,54]. Thus the current and earlier studies align with the notion that whole-cortex burst (or power) patterns in alpha (primarily in sensory and parietal regions) and beta (primarily in parietal and prefrontal regions) dynamically evolve to orchestrate the flow of sensory information to be stored in or deleted from WM according to behavioural (task) demands[10,48].

These cortical burst patterns are in turn likely coordinated in interactions with thalamus and basal ganglia[5,15,39,44,55–60]. This was supported by the observation that the differential beta bursting between distractor and no-distractor presentations observed in occipital cortex was mirrored in the thalamus and dorsal striatum in source reconstructed data. Interestingly however, prefrontal cortex tended to lead bursting in the sub-cortical structures. We also observed differential alpha and beta bursting for distractors in fusiform gyrus. Earlier studies implicated temporal cortex in WM, and fusiform gyrus specifically for face WM[61–63]. Recently the level of beta frequency coupling between prefrontal and inferotemporal cortex predicted performance on a WM task in non-human primates[64].

In summary, our results help unite the body of findings regarding the role of beta and alpha in attention and WM in different cortical areas, across human and non-human primates. They appear to serve distinct roles, which can be further teased apart by future experiments.

## Methods

### Participants

We recruited 19 participants, 12 males and 7 females, aged 21-41 years (mean 27.1), with no known cognitive impairments and tested with Ishihara's test for colour deficiency without remarks. The participants were primarily recruited students and received 500 SEK in compensation vouchers. One male did not perform the experiment due to metal interfering with the MEG scanner, and one male was excluded due to poor data quality resulting from excessive movement during the experiment resulting in a total of 17 participants. The Swedish Ethical Review Authority (Dnr 2021-00336) approved the study. All participants were thoroughly briefed and provided written consent to their participation.

### Experimental paradigm

The purpose of the task was to study burst dynamics across regions and frequencies during the presentation of stimuli and the effect of distractors. The task was a sequential working memory task with 4 foveally presented, randomly oriented bars (Fig. 1). Before the onset of each sequence there was a pre-cue indicating if the current trial was a Distractor (50%) or No-Distractor trial (50%). Distractor and No-Distractor trials were randomly interleaved with the criterion that a maximum of 3 trials of a certain type could be presented in a row. On No-Distractor trials the participants were tasked with remembering the orientation of all four bars. On Distractor trials only the first and the fourth bars were to be remembered, and the second and third bars were distractors. The No-Distractor trials were indicated by a pre-cue with four parallel vertical solid bars, while the Distractor trials were indicated by the same bars except that the second and third bars were illustrated with dashed edges informing the participant that these were distractors (not to be remembered). Except for the pre-cue, the two types of trials were visually identical, which allowed us to study the neural mechanisms of encoding, filtering and removing information by comparing a to-be memorized item with a distractor. Each trial commenced on a black background, with a white fixation dot during 500 ms, followed by a pre-cue shown for 500 ms. Next, four bars with random orientation, each covering a visual angle of 7.1° were shown sequentially for 500 ms, separated by a brief delay period of 500 ms with a centrally placed white fixation dot. Each individual bar in the sequence was marked with a uniquely coloured fixation dot in the centre (the colour was randomly selected on each trial from red, green, blue and magenta). Following the presentation of the fourth bar, a white fixation dot was shown for 750 ms. The fixation thereafter changed colour (red, green, blue or magenta) acting as a retro-cue. The retro-cue identified the upcoming bar which would be tested for 750 ms after which the subject was presented with a randomly rotated probe. Using a 4-button control pad, the subjects had 5 seconds to rotate the probe to match the memorized bar. The subjects had to perform 400 trials in blocks of 40 with pauses in between. In each pause the subject was asked to decide when to continue, and in total the task took between 60-70 minutes for the subjects to complete.

### Frequency Tagging

Frequency tagging involves manipulating the spectral power. This is achieved by oscillatory modulation of the intensity of the stimuli, which entrains neural activity to the frequency of the modulation. This provides insights into when and where the sensory stimuli are processed in the brain (as measured by MEG). Frequency tagging of the stimuli has been applied in both auditory[65,66] and visual perception[30,33]. We modulated the luminance of the cue and the four bars by a sine function phase-locked to the onset of the stimuli with tagging frequencies of either 31.1 or 37.1 Hz. We chose prime frequencies to avoid sub-harmonics, and the decimal was added from the result of a limited pilot study. Within each trial, the objects in the sequence were tagged at alternating frequencies (31.1 Hz and 37.1 Hz), with 50% of the trials tagging the first object with 31.1 Hz and the rest at 37.1 Hz. The stimuli with a frequency tag appeared as faint flickering and no visual difference between the two frequencies was seen or reported.

### Procedure, Materials and Data acquisition

Participants were sent information about the experiment in advance and upon arrival they were greeted and fully informed about the task. They signed a consent form, a MEG screening form and provided a suitable set of clothing to change into. All jewellery, hairpins and any other objects were removed at this stage. The participants were then prepared for the procedure by fitting electrodes and head digitalization. Next, they were led into a magnetically shielded room where the experiment was conducted. Before the MEG recordings were made, each subject was given time to practice the task for about 25-40 trials.

The MEG scanner was an Elekta Neuroma TRIUX 306-channel, located inside a 2-layer magnetically shielded room (www.natmeg.com). The procedure was presented on a FL35 LED DLP Projector from Projection Design running at 32 bit colour in 1920*1080 at 120 fps, and the experiment was performed using Presentation® software (Version 23.0, Neurobehavioural Systems, Inc., Berkeley, CA, www.neurobs.com). Eye movements and eye-blinks were recorded with an Eyelink 1000 binocular tracker from SR Research.

Data were recorded at the sampling rate of 1000 Hz for all 306 MEG channels, eye tracker channels, 16 event code channels and 2 channels dedicated to electrooculography and electrocardiography. Subjects submitted their responses using a 4-button inline pad from Current Design.

T1 MRI scans for all subjects were collected in a Siemens Prisma 3 Tesla whole-body MRI for source data analysis.

### Data pre-processing and frequency domain analysis

The recorded data files were run through Maxfilter software by Elekta Neuromag, with temporal signal space separation. Further analysis was done using Fieldtrip. Data files were segmented into trials starting one second before the trial start and one second after the submission of the response. Trials were demeaned, line noise removed and trials with jump artefacts, identified as a z-score greater than 80, were removed. This resulted in 15 trials removed on average per subject. Across the MEG channels 60 ICA components were calculated using FastICA and then used to automatically identify and remove EOG and muscle artefacts using procedures provided by Fieldtrip. ECG artefacts were identified and removed using a semi-automatic procedure as described by Fieldtrip. The data was down sampled to 250 Hz. Planar gradiometers were combined into synthetic sensors and grouped into regions of interests representing occipital, parietal and PFC regions (Figure S7).

Time frequency calculations were done for single trials in the range 5-45 Hz using adaptive, multiplicative superlets[67] with order

ranging from 1 to 10, and a base wavelet length of three cycles. Only data from planar gradiometers were used. Except for burst calculations, a trial-by-trial z-score baseline was calculated based on the 250 ms epoch before the pre-cue onset. Total power was calculated from the pre-processed activity data, and induced power was calculated by removing the ERF from the activity data and applying the same method. Phase-locked power was calculated by subtracting induced power from total power[68].

### Burst extraction

To identify bursts of high power, we first specified three frequency bands of interest: 8-12 Hz (alpha), 12-18 Hz (low beta), or 18-26 Hz (high beta). The cut-off at 26 Hz was chosen to avoid spectral leakage from the tagging frequency at 31.1 Hz. Within each band and for each sensor we calculated the temporal profile of the induced power using adaptive superlets. Bursts were defined as intervals within each trial where instantaneous power was above 1.5 standard deviations above the mean (running average of the last 10 full trials), and with the duration of at least two cycles of the average frequency of the band[8]. For each subject, the burst events were averaged across the stimuli into a burst rate per subject, from which a grand average burst rate (per frequency band and cortical region) was calculated. Individual burst times were also kept for further single trial analysis.

### Burst triggered frequency tagged power

Burst events for each frequency band, subject, channel and trial were extracted for a window of 200-700 ms after the onset of a frequency tagged stimuli. Each event was associated with its descriptive information (subject, channel, trial type), and the spectral power within the tagged frequency (narrow band of 4 Hz) was extracted from phase-locked power, during a window of 30 ms before and after the onset of the burst event. The burst triggered frequency tagged phase-locked power was calculated by averaging events within subject, and then across subjects.

### Statistical analyses

To analyse behaviour, we collected all responses and compared their angles to the angles of the probed bars. Analysis was done in in R using packages tidyr, dplyr, tidyverse and circstats. The circular error, defined as the angle of the response subtracted from the angle of the probed bar in a circular reference frame (from −90° to +90°), was calculated for all trials. We then took the absolute value of the error and applied a transformation, to assure that the assumptions of the general linear model were met. Zero errors were set to 0.5 degrees and we applied a Box-Cox transformation with parameter L = 0.101. The parameter L was estimated to meet as close as possible the normality assumptions. Posterior inspection of the residuals showed a good fit with the assumptions of normality (Q-Q plots) and homogeneity of the variances. Mixed effects models where fitted in R[69]. The following packages were used: lme4[70], lmerTest[71], Mass[72], performance[73].

To relate bursts rates between the tasks (Figs. 5, 6 and 7), cluster based non-parametric tests[74] were performed for the burst activity across subjects. Trial types were shuffled within each channel and subject in order to preserve the structure of the data and a grand average was calculated across subjects and channels and t-statistics were calculated and filtered at the 99.9% percentile in sensor space, and 99% percentile in source space. Clusters were defined as the sum of temporally adjacent t values and for each sample the largest cluster was extracted. The procedure was repeated 10,000 times generating a histogram of clusters at which a limit of 99.9% in sensor space, and 99% in source space, was used to identify significant clusters in the original data.

The test for burst frequency tagged power required a different approach (Fig. 8). A matrix containing all burst events with their tagging power and descriptive information was constructed, and trial numbers were shuffled 10,000 times within their descriptive groups. Averages were calculated and compared to actual data in order to calculate p-values, and tests were performed at the 99.9% percentile.

To compare burst events between source regions (Figure S4), cluster-based non-parametric tests were conducted to contrast burst rates in the target region before (0.3 s to onset) and after (onset to 0.3 s) the burst in the source region. Burst data in the target region, centred around the onset of the trigger in the source region, was divided into pre-burst and post-burst periods at the onset. The pre-burst data was mirrored along the time axis and shuffled with the post-burst data and 10,000 pairs of data were drawn and t-values calculated for each subject id, preserving the structure of the data. Two-sided cluster-based permutation test were performed on the t-values with a cluster threshold of 99%, and significance level of 99% contrasting pre-trigger bursts rates with post-trigger burst rates.

### Source Space Data Analysis

Source-level analysis was conducted using Fieldtrip[75]. Individual T1 MRI scans were co-registered with the MEG scanner, re-sliced, and segmented into brain, skull, and scalp volumes. The brain segment was used to construct a volume conduction head model. A spatial filter was calculated on a 10 mm grid, and individual anatomies were warped onto a standard MNI template. The spatial filter was employed to calculate trial-level activity data in source space using Minimum Norm Estimation[76]. Source-level activity was subsequently parcellated using the Anatomical Automatic Labelling (AAL) atlas[77]) into 116 volumes. Time-frequency calculations and burst extraction of the parcellated volumes was performed as described above.

Interregional burst correlations were calculated for each frequency band, subject, trial, and source region of interest. This involved extracting burst onset times from the originating region of interest. For each event, a symmetric time window of ±0.3 seconds was applied to detect burst onsets within the corresponding target region of interest during the same trial. The detected onsets were then aggregated across all subjects and trials. For visual representation in Figure S4, data preceding each burst event were mirrored along the y-axis to facilitate comparison and enhance interpretability. The volumes of interest used for parcellation using the AAL atlas were PFC (Frontal_Sup_L, Frontal_Sup_R, Frontal_Sup_Orb_L, Frontal_Sup_Orb_R, Frontal_Mid_L, Frontal_Mid_R, Frontal_Mid_Orb_L, Frontal_Mid_Orb_R, Frontal_Inf_Oper_L, Frontal_Inf_Oper_R, Frontal_Inf_Tri_L, Frontal_Inf_Tri_R, Frontal_Inf_Orb_L, Frontal_Inf_Orb_R, Frontal_Sup_Medial_L, Frontal_Sup_Medial_R, Frontal_Med_Orb_L, Frontal_Med_Orb_R), Parietal cortex (Parietal_Sup_L, Parietal_Sup_R, Parietal_Inf_L, Parietal_Inf_R), Occipital cortex (Calcarine_L, Calcarine_R, Cuneus_L, Cuneus_R, Occipital_Sup_L, Occipital_Sup_R, Occipital_Mid_L, Occipital_Mid_R, Occipital_Inf_L, Occipital_Inf_R, Precuneus_L, Precuneus_R, Basal Ganglia (Caudate_L, Caudate_R, Putamen_L, Putamen_R, Pallidum_L, Pallidum_R) and Thalamus (Thalamus_L, Thalamus_R).

### Reporting summary

Further information on research design is available in the Nature Portfolio Reporting Summary linked to this article.

## Data availability

The processed data, downsampled MEG data, MRI data and behavioural data used in this study are available in the OSF database under accession code CC BY (https://osf.io/gu25f/?view_only=6412a8bd665e4ef082385dbaa3d33026). Source data are provided with this paper.

## Code availability

The code for this study is available for download from the Open Science Framework (OSF) homepage (https://osf.io/gu25f/?view_only=6412a8bd665e4ef082385dbaa3d33026).

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

## Acknowledgements
This work was funded by ERC starting grant 949131, and Swedish Research Council (Vetenskapsrådet) grants 2018-04197, 2022-02328 (M.L.).

## Author contributions
Concept: M.L. Experimental design: M.L. and R.A. Data collection: J.Li and G.R. Methodology: M.L. and P.H. Software: P.H. Formal analysis: J.Li. and R.A. Supervision: J.Lu., M.L., R.A., and P.H. Visualization: J.L. Wrote first draft: M.L. Text edits: P.H., R.A., J.Lu., J.Li., and M.L. Resources: M.L.

## Funding

## Competing interests
The authors declare no competing interests.
