## [Peer Review File · Nature Communications]

Distinct Functions for Beta and Alpha Bursts in Gating of Human Working MemoryREVIEWER COMMENTS

Reviewer #1 (Remarks to the Author):

Liljefors et al. present an interesting MEG study that seeks to establish connections between previous findings on single-trial alpha and beta activity in working memory (WM) processing in humans and analogous studies conducted on non-human primates by the same research group. The study employs a sequential WM task, wherein participants are instructed to remember either all presented items (No-Distractor condition) or only 2 out of the 4 items (Distractor condition). This design allows the authors to compare single-trial alpha and beta burst activity for stimuli intended to be encoded and those meant to be ignored. Presented items are further frequency-tagged to track down item-specific WM processing. The results indicate that both alpha and beta bursts exhibit similar patterns to those reported in earlier non-human primate studies, suggesting the involvement of both frequency bands in inhibitory control of WM processing. The authors highlight distinct differences in the frequency bands concerning brain region and processing time, indicating that alpha and beta activity play similar roles with nuanced differences.

Although the study utilizes a well-designed approach, robust methodologies, and addresses a timely and intriguing research question, the results seem somewhat incremental. In the present version, differentiating activity related to WM processing from that linked to visual attention proves challenging, and the authors do not fully capitalize on the single-trial analysis to explore the intricacies of successful WM content processing and retention. Furthermore, the lack of a connection between the reported brain dynamics and behavioral performance diminishes the potential impact of the findings on the field. The subsequent section outlines several questions and concerns in no particular order.

1. The single-trial burst activity analysis enables the authors to relate recorded brain dynamics to successful WM performance on a trial-by-trial basis – one of the key advantages of their analysis approach. Yet, most of the findings presented are based on analysis of trial-averaged burst activity. The findings therefore fall short of providing much deeper insights than what has been shown by trial-averaged power analyses in many earlier MEG and EEG studies. The only analysis for which the authors utilize the single-trial approach is the burst-triggered tag-power analysis. The presented results from this analysis, however, are rather weak and the specificity of the results remain unclear (more below). In my opinion, the relation between burst activity and WM processing (and thus also the impact of the study) could be much enhanced by establishing a link between single-trial burst activity and WM accuracy and performance. Since the authors measure WM accuracy in a continuous fashion for each single trial (degrees distance to original rotation), they are well-positioned to analyze the effect of single-trial burst activity on single-trial WM accuracy. This could yield much deeper insights into the mechanisms of WM processing and inhibitory control.

2. On a related note, due to the missing link to WM performance, it is currently difficult to relate the findings to WM processing in general. Most of the findings – if not all – could also be related to differences in attentional demands and temporal expectations between the two conditions. Since in distractor trials, participants know that the second and third item will always be a distractor item, they will by design pay much less attention to these items. The enhanced alpha and beta power before, during, and after presentation of the distractor items are well in line with inhibitory roles both frequency bands have in visual attention (i.e., ‘gating-by-inhibition’ (Jensen & Mazaheri, 2010)). Once again, it would be highly valuable to establish a relationship between recorded brain activity and single-trial WM performance, providing a more nuanced understanding of WM processing distinct from attentional demands.

3. A consistent finding is the increased burst rate of alpha and beta oscillations right before the presentation of the retrocue across all areas. The authors interpret this as indicative of load differences between conditions. However, if this were the case, load differences should be evident with the presentation of the second and third items since in one condition, load increases, while in the other, it remains constant. How do the authors explain the strong difference only right before the retrocue? Is that still in line with an inhibitory role for alpha and beta? Further elaboration on this point would be appreciated.

4. The authors report strongest fluctuations of beta and alpha bursts in occipital and parietal areas. Fluctuations are also present in pre-frontal areas but there are much weaker. This contrasts with results in non-human primates where robust fluctuations were observed in pre-frontal areas. How do the authors explain these discrepancies between the species?

5. Enhanced insights into the involved brain regions could be achieved through source-level analyses and statistics. Currently, sensor-level activity is averaged in three regions of interest (it is unclear which channels were used for each area, unless I missed this info), making interpretability challenging due to the linear combination of all underlying sources. It would be insightful to perform source projections and condition contrasts in source space to reveal more nuanced differences in the brain areas involved.

6. The frequency tagging of WM items is an intriguing design choice, yet the authors appear to underutilize it. While they demonstrate weaker evoked power in the tagged frequencies for distractor items, the significance of this effect is unclear (see below). Additionally, it would be beneficial to include a control to test whether similar effects can be observed in unrelated frequency bands. If bursts indeed have an inhibitory effect on a specific item in WM, they should see effects only for the tag-power in the frequency range of the current item (say 31Hz if used for that item), but not the other frequency (37 Hz). This would be a more valuable approach than

averaging across both frequency tags. Moreover, if inhibitory control is particularly a function of frontal and parietal areas, wouldn't it be more interesting to assess the tag power (in occ. channels) triggered by the onset of frontal/parietal bursts?

7. Figure 3 seems not to provide statistics. Which of the reported differences between the conditions are statistically significant in these plots? In general, it would be good to provide statistics also in the main text, not just in the figures. Otherwise, it is difficult to assess the significance of the results.

8. The authors chose to shuffle the color of each WM item per trial, introducing the need of participants to memorize not only the orientation of the bar but also the correct order of the color dots. This could result in an increased likelihood of "swap errors." Distinguishing between swap errors and recall errors (see Bays 2016, Sci Rep), could provide valuable insights when relating brain activity to WM performance.

Reviewer #2 (Remarks to the Author):

Information is selectively gated into and out of working memory in a goal-directed manner. Previous work has identified several spectral correlates of gating. In local field potentials recorded from primate prefrontal cortex, bursts of power in the beta band are reduced during encoding and active maintenance of task-relevant stimuli. Additionally, human EEG and MEG studies have identified alpha power as a correlate of filtering out task-irrelevant stimuli. In the present work, the authors aim to relate these largely distinct literatures by systematically examining alpha and beta across occipital, parietal, and prefrontal cortex during gating within a single study.

To do this, the authors recorded the magnetoencephalogram from human subjects performing a working memory task in which stimuli were pre-cued as either task-relevant ('targets') or task-irrelevant ('distractors'). Behavioral analyses and a clever examination of frequency-tagged stimulus evoked responses convincingly demonstrate that subjects ignored the distractors to improve their performance. The authors examine alpha and beta across cortex and report four principle findings: (1) In occipital cortex, alpha and beta bursts increased following distractor onset and were associated with reduced stimulus-evoked responses. (2) In parietal and prefrontal cortex, beta band bursting decreased during delays prior to distractor onset. (3) In parietal/prefrontal cortex, beta bursts decreased over the course of the trial. (4) Alpha and beta burst rate increased with load across all electrode sites prior to the onset of a retro-cue at the end of the trial. Based on these results, the authors hypothesize that alpha is associated with gating sensory signals while beta is associated with gating into and out of working memory. These results will be of interest to

researchers studying working memory in a range of model systems and should inspire further experimental and theoretical work interrogating the mechanistic basis of these signals.

Major Comments:

1. The study is well-designed and the analyses that are reported are appropriate. However, statistical analyses are presented for only one of the four findings outlined above (#3). Given the stated goal of systematically examining alpha and beta across cortex, statistical analyses of how each phenomenon of interest is modulated by region, frequency band, and their interaction would significantly improve the rigor of the paper.

2. On a related note, it would be helpful to visualize the 'average distractor vs target response' and the 'average delay period preceding distractors' (Fig. 6) for each of the three ROIs, so that the reader can assess their cortical specificity.

3. Figure 3B is never called out explicitly in the text — could the authors walk the reader through the key takeaway(s)? Similarly, differences in induced power between distractors and no-distractor trials are presented in Figure 3A and briefly alluded to in the text, but never unpacked.

4. The authors raise the interesting possibility in the introduction that some of the discrepancies in the literature around the role alpha and beta in gating could be due to analysis methods (power fluctuations or bursts). In these data, would analysis of power fluctuations rather than burst rate yield qualitatively different results?

Minor Comments:

1. Previous work from the authors defined beta as 20-35 Hz. Some language in the methods describing how the beta bands (12-18 and 18-26) were selected and what motivated the low/high split would be helpful.

2. Line 198: in this first sentence, did the authors mean to highlight the distractor/no-distractor *difference* just prior to the retro-cue?

Reviewer #3 (Remarks to the Author):

Liljefors and colleagues show that alpha and beta bursts are differently modulated during a sequential working memory task in humans. They argue that alpha bursts are involved in the suppression of unwanted stimuli, given an increased reduction of alpha bursts during target versus distractor processing. Beta bursts are argued to be involved in the transition from stimulus processing to WM retention, based on temporal modulations of the bursts in response to stimulus presentations, and are also argued to be involved in the proactive suppression of distractor stimuli, based on an increase in beta bursts before distractor presentation.

This paper presents compelling and relatively straightforward results, that link to previous work in non-human primates. However, I have some concerns which I will elaborate on in more detail below:

The interpretation of the results need to acknowledge that these results may not be specific to working memory (or explain better why they think it is). Also the wider relevant literature concerning alpha and beta power modulations in humans needs to be acknowledged.

The stimulus evoked reduction in alpha and beta power is a well-known effect, often referred to as event-related-desynchronisation (ERD; Pfurtscheller & Aranibar, 1977). This phenomenon has been widely studied, and is modulated by attention and stimulus saliency. The observed results (ERD in general and more ERD for targets versus distractors) is thus not surprising, and not specific to working memory.

The authors tend to, in particular in the abstract, imply a causal role between alpha and beta bursts and working memory mechanisms. In my opinion, this is unwarranted and should be toned down. What has been found is a relationship between working memory and the measured neural responses. This does not mean that alpha and beta bursts actually regulate working memory processes.

The authors report increased beta power bursts before target versus distractor trials and interpret this as evidence for “down-regulation of existing WM-related activity”. However, WM-specificity was not tested here since the distractors were always presented after a target stimulus. A comparison to distractor trials without a preceding target stimulus would be necessary to make such claims. The effect reported here could be related to preparatory attention (getting ready to encode a target, versus ignoring a distractor).

The authors use, what they call “bursts” as their main measurement. This needs to be better explained. Why exactly is this method used over the more conventional total power? And how does it differ? I fail to see any advantage or even notable differences for this metric. To be clear, I am not saying that there aren’t any, but if there are, the authors need to better highlight and explain those. Would it be possible to also do the analyses on total power (Figures 4, 5, 6)? Then the results can be more easily compared with the wider literature, while highlighting the potential differences between bursts and power.

Minor comments:

In lines 398-399 it is stated "For analysis for which a baseline was applied..." However, as far as I could tell, it is not mentioned again which analyses used a baseline.

Figure 1: The last delay, after the target 4 is 0.5 s in duration. The figure caption and the methods state that it is 750ms, however.

Reviewer #3 (Remarks on code availability):

The scripts to reproduce figures seem to be all there, as well as a readme file with instructions. However, no data has been uploaded, so the scripts cannot be run.

REVIEWER COMMENTS

We would like to thank the reviewers for taking their time to provide us with constructive critique. To address the reviewers' concerns we have performed several additional analyses including behavioral-neural correlations, source reconstruction, additional control analysis and power analysis. These analyses revealed interesting relationships between bursts across regions, between bursts and behavior and also helped us relate our original findings to classical power analysis.

It has led to the addition of 7 new figures (and some minor changes to existing ones), six of which are collected in a new supplemental material (Figure S4 contains interesting and novel results on burst timings between cortical regions and could be changed into a main figure if the Editor/Reviewers prefer). These analyses support our initial results, and also led to novel findings. We have in addition rewritten substantial portions of the MS (tracked) and added a lengthier discussion of the existing literature. Taken together we believe this has improved and drastically deepened the MS.

Reviewer #1 (Remarks to the Author):

Liljefors et al. present an interesting MEG study that seeks to establish connections between previous findings on single-trial alpha and beta activity in working memory (WM) processing in humans and analogous studies conducted on non-human primates by the same research group. The study employs a sequential WM task, wherein participants are instructed to remember either all presented items (No-Distractor condition) or only 2 out of the 4 items (Distractor condition). This design allows the authors to compare single-trial alpha and beta burst activity for stimuli intended to be encoded and those meant to be ignored. Presented items are further frequency-tagged to track down item-specific WM processing. The results indicate that both alpha and beta bursts exhibit similar patterns to those reported in earlier non-human primate studies, suggesting the involvement of both frequency bands in inhibitory control of WM processing. The authors highlight distinct differences in the frequency bands concerning brain region and processing time, indicating that alpha and beta activity play similar roles with nuanced differences.

Although the study utilizes a well-designed approach, robust methodologies, and addresses

a timely and intriguing research question, the results seem somewhat incremental. In the present version, differentiating activity related to WM processing from that linked to visual attention proves challenging, and the authors do not fully capitalize on the single-trial analysis to explore the intricacies of successful WM content processing and retention. Furthermore, the lack of a connection between the reported brain dynamics and behavioral performance diminishes the potential impact of the findings on the field. The subsequent section outlines several questions and concerns in no particular order.

Thank you for the detailed and constructive feedback. WM and attention are closely related processes (selecting a particular item held in WM is often likened to internal attention), and the two concepts largely but not fully share neural mechanisms. We agree that in particular a link between behavior and brain dynamics was missing, and we have now addressed this issue together with other helpful suggestions from the reviewer. We now present much more detailed single trial analysis rather than proof of concept. We present both how it links to behavior and intricacies in electrophysiology during encoding and retention.

1. The single-trial burst activity analysis enables the authors to relate recorded brain dynamics to successful WM performance on a trial-by-trial basis – one of the key advantages of their analysis approach. Yet, most of the findings presented are based on analysis of trial-averaged burst activity. The findings therefore fall short of providing much deeper insights than what has been shown by trial-averaged power analyses in many earlier MEG and EEG studies. The only analysis for which the authors utilize the single-trial approach is the burst-triggered tag-power analysis. The presented results from this analysis, however, are rather weak and the specificity of the results remain unclear (more below). In my opinion, the relation between burst activity and WM processing (and thus also the impact of the study) could be much enhanced by establishing a link between single-trial burst activity and WM accuracy and performance. Since the authors measure WM accuracy in a continuous fashion for each single trial (degrees distance to original rotation), they are well-positioned to analyze the effect of single-trial burst activity on single-trial WM accuracy. This could yield much deeper insights into the mechanisms of WM processing and inhibitory control.

We fully agree that we underutilized this aspect of the data, a decision we initially made to err on the more conservative side in our analyses. However, our primary goal was to establish a link with our earlier primate data (which we believe warrants also the single trial analysis).

Our initial thinking and observations were also that the WM behaviour was very complex, where the complete history of orientations of the bars in a given trial effected how subjects reported on a probed bar (similar to what is observed here; Bae et al., 2024). This reduced our hopes of being able to tie bursting at a given time in the trial to performance on a given bar. However, per the suggestion of the reviewer we explored this and found a salient link, now described in the manuscript. Our new finding connects beta and alpha bursting to performance on a single trial level (in the section "Beta and alpha bursts correlated with encoding on single trials"). In short, we built a linear mixed model taking into account the order and status of the probed item, as well as bursting around its encoding (trial type, target and burst rates as explanatory variables, using WM accuracy as the output variable).

We now write:

"Finally, to relate bursts to behaviour in single trials, we constructed a linear mixed effects model when targets were at the first or last bar. The performance data was transformed in order to meet the general linear model assumptions (see Methods). The model had subject as random effect, condition, serial order of probed item, and burst count as fixed effects. We performed this analysis for each frequency band separately. It suggested significant relationships in all frequency bands (Alpha burst count ($t=4.18$, $p=3e-5$, $df=4908$), probed item order ($t = -15.87$, $p < 2e-16$, $df=4896$), trial type ($t = 8.35$, $p < 2e-16$, $df=4896$); Low beta burst count ($t=3.42$, $p=0.0006$, $df=4896$), probed item order ($t = -15.67$, $p < 2e-16$, $df=4896$), trial type ($t = 8.37$, $p < 2e-16$, $df=4896$); high beta burst count ($t=2.71$, $p=0.0068$, $df=4896$), probed item order ($t = -15.06$, $p < 2e-16$, $df=4896$), trial type ($t = 8.39$, $p < 2e-16$, $df=4896$)), such that higher burst counts during distractors subsequently led to higher WM accuracy. In addition, we performed the same analysis but using burst counts in prefrontal and parietal regions preceding the second and third stimuli. It suggested weak evidence in the high beta band which was not significant (Alpha burst count: $t=1.16$, $p=0.25$, $df=4910$; Low beta burst count: $t=1.37$, $p=0.17$, $df=4906$; High beta burst count: $t=1.89$, $p=0.06$, $df=4901$). "

In addition, we utilize the single trial analysis to study timing of bursts across brain areas during WM encoding and retention. This yielded robust and informative patterns that also shifted from encoding to retention (Figure S4).

Bae, G. Y. (2024). Cardinal bias interacts with the stimulus history bias in orientation working memory. Attention, Perception, & Psychophysics, 1-10.

2. On a related note, due to the missing link to WM performance, it is currently difficult to

relate the findings to WM processing in general. Most of the findings – if not all – could also be related to differences in attentional demands and temporal expectations between the two conditions. Since in distractor trials, participants know that the second and third item will always be a distractor item, they will by design pay much less attention to these items. The enhanced alpha and beta power before, during, and after presentation of the distractor items are well in line with inhibitory roles both frequency bands have in visual attention (i.e., ‘gating-by-inhibition’ (Jensen & Mazaheri, 2010)). Once again, it would be highly valuable to establish a relationship between recorded brain activity and single-trial WM performance, providing a more nuanced understanding of WM processing distinct from attentional demands.

We fully acknowledge that neural mechanisms between attention and WM are often shared or partially overlapping. As presented in our response to R1, major concern 1, we now tie the alpha and beta filtering to the WM performance on a single trial level (and also refer to the “gating-by-inhibition” literature).

3. A consistent finding is the increased burst rate of alpha and beta oscillations right before the presentation of the retrocue across all areas. The authors interpret this as indicative of load differences between conditions. However, if this were the case, load differences should be evident with the presentation of the second and third items since in one condition, load increases, while in the other, it remains constant. How do the authors explain the strong difference only right before the retrocue? Is that still in line with an inhibitory role for alpha and beta? Further elaboration on this point would be appreciated.

Thank you. We had actually failed to appreciate this inconsistency. It is not easy to explain regardless of whether one assigns an inhibitory or excitatory role to alpha and beta. While we believe the load effect at the final delay to be consistent (this has also been presented by others in terms of alpha power and interpreted as inhibition/protection of WM), it is the lack of any load effect before the last delay which is puzzling. We now discuss this and provide potential explanations;

We now write:

“We also observed a somewhat puzzling finding from the perspective of beta and alpha bursting providing transient inhibition. While at the end of the delay, the retro-cue induced

more alpha and beta bursting in no-distractor trials, consistent with more inhibition when more items need to be suppressed around the retro-cue, or protected from sensory interference, there were no load effects prior to the final delay (i.e. between distractor and no distractor trials). It could potentially be that alpha and beta bursting during the encoding sequence mainly reflected executive control for allocation of resources, that in total were similar in the two sets of trials (with more resources per item and better performance in load 2 trials). The consistent reduction in beta bursting observed in the parietal and prefrontal sensors (and sources) with the presentation of each item regardless of cognitive status might reflect a mechanism of tracking the sequence of events during the trial, evident in both conditions to an equal degree. Prior studies observed load dependent alpha and beta power during retention, but have then not analyzed the gradual buildup of load during the encoding phase (Jensen et al., 2002; Kornblith et al., 2016)”

Jensen, O., Gelfand, J., Kounios, J., & Lisman, J. E. (2002). Oscillations in the alpha band (9–12 Hz) increase with memory load during retention in a short-term memory task. Cerebral cortex, 12(8), 877-882.

Kornblith, S., Buschman, T. J., & Miller, E. K. (2016). Stimulus load and oscillatory activity in higher cortex. Cerebral Cortex, 26(9), 3772-3784.

4. The authors report strongest fluctuations of beta and alpha bursts in occipital and parietal areas. Fluctuations are also present in pre-frontal areas but there are much weaker. This contrasts with results in non-human primates where robust fluctuations were observed in pre-frontal areas. How do the authors explain these discrepancies between the species?

We attribute these differences (apart from potential genuine difference between species) to two factors. First, the signal to noise ratio was much lower in frontal sensors compared to the occipital and parietal regions, due to the nature of MEG where sensors tends to be closer to occipital than frontal regions (which was not directly on the scalp due to the solid helmet-like setup). Thus either intracranial recording, or on scalp MEG may help resolve this. Second, a clear difference between non-human primate data and human data is that human subjects typically record one or two sessions, whereas non-human primates are exposed to the same task over the course of months. This seems to create stronger item dependent activation of prefrontal cortex also in humans when they are extensively trained (Miller, J. A., Tambini, A., Kiyonaga, A., & D'Esposito, M. (2022). Long-term learning transforms prefrontal cortex representations during working memory. Neuron, 110(22), 3805-3819). Thus this difference in training leading up to recordings may explain the differential focus (which is quite common in

the literature) in the two species. We now added a section in Discussion:

“While we observed task-specific modulation of alpha and beta bursting in most of the brain, the patterns were relatively weak in prefrontal regions, at odds with the strong focus on prefrontal cortex in non-human primate WM (Goldman-Rakic, 1995; Miller et al., 2018). This likely stems from the nature of MEG recordings with relatively low signal to noise ratio over frontal regions. However, the timing of alpha and beta bursts in prefrontal regions tended to precede bursts in other areas, including subcortical structures, suggesting a key role for prefrontal bursting. Additionally, a recent human study suggested that persistent training over time, as is typically the case in non-human primates, make prefrontal WM representations stronger, providing a potential explanation for the differential focus on brain areas between species (Miller et al., 2022).”

5. Enhanced insights into the involved brain regions could be achieved through source-level analyses and statistics. Currently, sensor-level activity is averaged in three regions of interest (it is unclear which channels were used for each area, unless I missed this info), making interpretability challenging due to the linear combination of all underlying sources. It would be insightful to perform source projections and condition contrasts in source space to reveal more nuanced differences in the brain areas involved.

We agree source level analysis would add additional important information. We initially opted to not perform this type of analyses given that sensor level analysis rendered very robust results (and thus should be highly reproducible with minimal processing). Thus to be as general as possible, and because we had such distinct effects between occipital, parietal and prefrontal sensors, we focused on this analysis. We do agree, however that a description of sensors included to actually promote reproducibility was missing. It is now added in Methods, with a new supplemental Figure (Figure S7) showing the electrode layout and which were used for each region.

In addition, we have now added source level analysis (New section “Source reconstruction suggested the involvement of sub-cortical structures”, with one figure and two supplemental figures). It revealed results that were consistent with the sensor level analysis, but also provided additional information. One important finding was that the burst filtering was localized to occipital rather than the occipito-parietal regions that has been suggested by previous studies on attention (and thus more in line with a directly inhibitory function for alpha/beta than those studies). Moreover, it suggested that thalamic and subthalamic regions

were involved in the filtering of distractors. It also suggested that prefrontal bursting tended to precede that of other cortical and subcortical areas. Overall, we definitely agree with the reviewer that these suggested analysis on source level contributed additional and valuable insights.

6. The frequency tagging of WM items is an intriguing design choice, yet the authors appear to underutilize it. While they demonstrate weaker evoked power in the tagged frequencies for distractor items, the significance of this effect is unclear (see below). Additionally, it would be beneficial to include a control to test whether similar effects can be observed in unrelated frequency bands. If bursts indeed have an inhibitory effect on a specific item in WM, they should see effects only for the tag-power in the frequency range of the current item (say 31Hz if used for that item), but not the other frequency (37 Hz). This would be a more valuable approach than averaging across both frequency tags. Moreover, if inhibitory control is particularly a function of frontal and parietal areas, wouldn't it be more interesting to assess the tag power (in occ. channels) triggered by the onset of frontal/parietal bursts?

We agree that a control analysis to test the specificity of the results would add key insights and gravitas to the analysis. Unfortunately, the frequency tag in 37 Hz also evoked some power around 31 Hz, and thus we initially did not perform such analysis. However, it is still true that the effects, while qualitatively similar should be much weaker when studying the non-tagged frequency. This is also indeed what we observe with an order of magnitude weaker effect in the non-tagged frequency (Figure S6). Moreover, we now display the effects for each tagged frequency separately in supplemental material (Figure S5) to visually show the interested reader that they are similar.

Second, we do not think or promote that alpha and beta is primarily a function of parietal and prefrontal areas, merely that it has distinct function in sensory vs higher order areas. In the case of sensory filtering of unwanted stimuli we believe that this should indeed be attributed to occipital bursts where sensory inputs are first received (along the lines of the current analysis, specifically the new source level analysis).

7. Figure 3 seems not to provide statistics. Which of the reported differences between the conditions are statistically significant in these plots? In general, it would be good to provide statistics also in the main text, not just in the figures. Otherwise, it is difficult to assess the significance of the results.

We now report statistically significant clusters in Figure 3. In addition, since we (mostly) used cluster-based permutation tests we want avoid referring to their precise extent in the text (since precise timings cannot be inferred; Sassenhagen and Draschkow, 2019). Thus we have not added description of significant clusters (in terms of exact frequency in Figure 3, or onset/offset in time) but have now provided p-values for the significant clusters in the text, and also refer to the relevant figure panels showing the specific effects we are discussing for easier identification throughout the main text.

*Sassenhagen, J., & Draschkow, D. (2019). Cluster-based permutation tests of MEG/EEG data do not establish significance of effect latency or location. *Psychophysiology*, 56(6), e13335.*

8. The authors chose to shuffle the color of each WM item per trial, introducing the need of participants to memorize not only the orientation of the bar but also the correct order of the color dots. This could result in an increased likelihood of "swap errors." Distinguishing between swap errors and recall errors (see Bays 2016, Sci Rep), could provide valuable insights when relating brain activity to WM performance.

We only partially agree with this comment. By shuffling the colors subjects actually did not have to keep track of order of items (colors were enough). However, swap errors (along with other biases discussed above) are still likely to occur. Thus, per the reviewer's suggestion, we analyzed data using a model that included potential swap errors. Indeed, we observed evidence for swap errors (see figure below). We did not include swap error analysis in the correlations with burst rates however, because it is difficult to assess on a single trial level whether a swap error occurred. We therefore did not include this analysis in the MS.

*Histograms of reported orientations, as angular distance to the bars that were not probed. Non-uniform distributions (tested with Kolmogorov/Smirnov tests and marked by *) indicate evidence of swap errors where subjects reported the orientation of a non-probed bar. This occurred for all targets except the last (helping to explain why accuracy was higher on the last bar).*

Reviewer #2 (Remarks to the Author):

Information is selectively gated into and out of working memory in a goal-directed manner. Previous work has identified several spectral correlates of gating. In local field potentials recorded from primate prefrontal cortex, bursts of power in the beta band are reduced during encoding and active maintenance of task-relevant stimuli. Additionally, human EEG and MEG studies have identified alpha power as a correlate of filtering out task-irrelevant stimuli. In the present work, the authors aim to relate these largely distinct literatures by systematically examining alpha and beta across occipital, parietal, and prefrontal cortex during gating within a single study.

To do this, the authors recorded the magnetoencephalogram from human subjects performing a working memory task in which stimuli were pre-cued as either task-relevant ('targets') or task-irrelevant ('distractors'). Behavioral analyses and a clever examination of frequency-tagged stimulus evoked responses convincingly demonstrate that subjects ignored the distractors to improve their performance. The authors examine alpha and beta across cortex and report four principle findings: **(1) In occipital cortex, alpha and beta bursts increased following distractor onset and were associated with reduced stimulus-evoked responses. (2) In parietal and prefrontal cortex, beta band bursting decreased during delays prior to distractor onset. (3) In parietal/prefrontal cortex, beta bursts decreased over the course of the trial. (4) Alpha and beta burst rate increased with load across all electrode sites prior to the onset of a retro-cue at the end of the trial.** Based on these results, the authors hypothesize that alpha is associated with gating sensory signals while beta is associated with gating into and out of working memory. These results will be of interest to researchers studying working memory in a range of model systems and should inspire further experimental and theoretical work interrogating the mechanistic basis of these signals.

Thank you for the kind and accurate summary of our findings and encouraging words. We agree and hope that the findings will inspire future studies on WM gating.

Major Comments:

1. The study is well-designed and the analyses that are reported are appropriate. However, statistical analyses are presented for only one of the four findings outlined above (#3). Given the stated goal of systematically examining alpha and beta across cortex, statistical analyses of how each phenomenon of interest is modulated by region, frequency band, and their interaction would significantly improve the rigor of the paper.

We already performed most of the tests asked for, but they were typically displayed in the figures only since they were cluster-based permutation tests. Describing their exact onset and offset in time in the text may cause readers to think these exact timings are statistically tested while they are not, and it is not considered good practice to do so (Sassenhagen and Draschkow, 2019). However, seeing that both Reviewer 1 (concern 7) and Reviewer 2 expected these comparisons to be in the text, we have added p-values in the text and made direct references to the figure panels displaying the clusters we are discussing throughout the MS.

Sassenhagen, J., & Draschkow, D. (2019). Cluster-based permutation tests of MEG/EEG data do not establish significance of effect latency or location. *Psychophysiology*, 56(6), e13335.

2. On a related note, it would be helpful to visualize the 'average distractor vs target response' and the 'average delay period preceding distractors' (Fig. 6) for each of the three ROIs, so that the reader can assess their cortical specificity.

We have now included this in a supplemental figure (Figure S2).

3. Figure 3B is never called out explicitly in the text — could the authors walk the reader through the key takeaway(s)? Similarly, differences in induced power between distractors and no-distractor trials are presented in Figure 3A and briefly alluded to in the text, but never unpacked.

We agree that description and analysis pertaining to Figure 3 was insufficiently described and linked to. We have now added a more in-depth description of these findings as well as statistics within the figure itself. We also removed parts of Figure 3B that was not central and better described later with the burst analysis:

"Because the frequency tagging was phase-locked to the stimulus onset, we analyzed total, phase-locked, and induced power separately. Neural activity corresponding to processing of the tagged stimuli would be most strongly observed in the phase-locked power as the tagging itself was phase-locked to the onset of stimulus (Figure 3a). We assessed neural substrates of gating by contrasting power in the No-Distractor and Distractor trials. For phase-locked power this analysis demonstrated that distractors entrained cortical activity in the tagged frequencies to a significantly lesser degree than targets around stimulus presentations (Figure 3a). The topography of these effects suggested the main difference in entrainment of tagged frequencies were around the occipital regions (Figure 3b). In induced power, instead, we observed significant differences in the alpha and beta frequency ranges between these two conditions around the same time (Figure 3a). Thus it suggested the involvement of alpha and beta oscillations as correlates of filtering information into WM."

4. The authors raise the interesting possibility in the introduction that some of the discrepancies in the literature around the role alpha and beta in gating could be due to analysis methods (power fluctuations or bursts). In these data, would analysis of power

fluctuations rather than burst rate yield qualitatively different results?

We have added a new figure (Supplemental Figure S1) demonstrating key findings replicated using power rather than bursts. The major difference is the increased sensitivity of burst analysis (apart from letting us do single trial analysis with tag power, timings or the conceptual aspects of bursts vs power). We argue that power is driven by burst events so overall power and burst rates should be highly correlated. Some subtle differences we observe, for instance that the different frequency bands are more similar in power, suggesting that the burst analysis helps us isolate differences (that may be obscured by bleeding in the power analysis. Bleeding contributes low power events in other bands that arguably are removed when focusing on high power bursts). As an example, the difference between alpha and beta in occipital regions between onset and offset of stimuli is not evident in power.

Minor Comments:

1. Previous work from the authors defined beta as 20-35 Hz. Some language in the methods describing how the beta bands (12-18 and 18-26) were selected and what motivated the low/high split would be helpful.

We agree with this discrepancy. It was motivated by our recent study which spanned not only prefrontal, but also parietal and occipital cortex in non-human primates (Lundqvist et al., 2020). It turned out that in other regions than prefrontal cortex beta tends to have a lower frequency, all the way down to typical alpha in occipital regions. It was to be able to pick up on these potential differences also in humans that we did the split. Upwards in frequency we were limited by the highest possible tag frequency (going to higher tag frequency created subharmonics). During piloting we also observed that alpha/beta tended to be somewhat lower in frequency in MEG in humans compared to intracranial recordings in primates. So this combined led to the choice of frequency bands. It is now motivated in Results:

"The choice of the distinct frequency bands was motivated by our recent findings in non-human primates that the "functional" beta frequency tend to be lower further down in the cortical hierarchy (Lundqvist et al. 2020). Thus there may be important differences between sub-bands in terms of cortical origin and function. Upwards we were bounded to avoid overlap with the tagging frequencies."

Note also that we changed how we refer to our earlier studies on beta to 14-35 Hz. This since we mostly have used 20-35 Hz in our prefrontal analysis but were going all the way down to 14 Hz in one of the papers cited (Lundqvist et al., 2020).

2. Line 198: in this first sentence, did the authors mean to highlight the distractor/no-distractor *difference* just prior to the retro-cue?

Yes, thank you, we meant to make a contrast between the two conditions in this statement, this has now been fixed.

"Finally, in the delay period following the sequence, just prior to the retro-cue there was a larger elevation of bursts for No-Distractor trials in all three frequency bands and all sensors."

Reviewer #3 (Remarks to the Author):

Liljefors and colleagues show that alpha and beta bursts are differently modulated during a sequential working memory task in humans. They argue that alpha bursts are involved in the suppression of unwanted stimuli, given an increased reduction of alpha bursts during target versus distractor processing. Beta bursts are argued to be involved in the transition from stimulus processing to WM retention, based on temporal modulations of the bursts in response to stimulus presentations, and are also argued to be involved in the proactive suppression of distractor stimuli, based on an increase in beta bursts before distractor presentation.

This paper presents compelling and relatively straightforward results, that link to previous work in non-human primates. However, I have some concerns which I will elaborate on in more detail below:

Thank you, we are happy about this generally positive assessment of our findings and mostly agree with the criticism.

The interpretation of the results need to acknowledge that these results may not be specific to working memory (or explain better why they think it is). Also the wider relevant literature concerning alpha and beta power modulations in humans needs to be acknowledged.

The stimulus evoked reduction in alpha and beta power is a well-known effect, often referred to as event-related-desynchronisation (ERD; Pfurtscheller & Aranibar, 1977). This phenomenon has been widely studied, and is modulated by attention and stimulus saliency. The observed results (ERD in general and more ERD for targets versus distractors) is thus not surprising, and not specific to working memory.

We agree that suppression/ERD of alpha/beta is not exclusive to WM. We thought we highlighted this point, especially in regards to alpha ERD in relation to the phenomenon of attention. But we agree ERD goes beyond attention and also is related to for instance storage of long term memory. Thus it is important to point out that our findings fit within this general scheme. We now discuss several studies on alpha/beta ERD. Primarily in the Discussion with a dedicated paragraph where we try to relate our findings to this literature, proposing a more general role for beta ERD in stimulus processing.

We now write:

“Changes in alpha and beta power during stimulus processing has been extensively studied (Pfurtscheller and Aranibar 1977; Klimesch et al. 2007; Jensen and Mazaheri 2010; Worden et al. 2000). It has been linked to encoding of information into episodic and long-term memory (Hanslmayr et al. 2014; Griffiths et al. 2021), and several theories suggest a link between alpha and the dynamic, functional architecture of cortex (Bastos et al. 2020; Jensen and Mazaheri 2010; Klimesch et al. 2007). Specifically related to our findings, human studies have suggested that modulation of alpha power reflects filtering of sensory processing in attention and WM tasks (Worden et al. 2000; Sauseng et al. 2005; Popov et al. 2017; Zhou et al. 2023; Zhigalov and Jensen 2020; Roux and Uhlhaas 2014; Haegens et al. 2012; Turner et al. 2023), although not necessarily under strict top-down control (Gutteling et al. 2022; Jensen 2024). Intracranial recordings demonstrate that alpha activity is finely tuned spatially, consistent with a role of selective suppression of unwanted information in a visual scene (Popov et al., 2019; Yuasa et al., 2023). Alpha power has also been reported to reflect the modality-specific suppression of distractors in in a WM task with visual and auditory inputs (Zhou et al., 2023). Alpha power has also been demonstrated to be high and upregulated in a load dependent manner during WM retention, consistent with a role in protecting WM information from distractors (Klimesch et al. 1999; Jensen et al. 2002; Bonnefond and Jensen 2012). Thus the current and earlier are in line with that whole-cortex burst (or power) patterns in alpha (primarily in sensory and parietal regions) and beta (primarily in parietal and prefrontal regions) dynamically evolve to orchestrate the flow of sensory information to be stored in or

deleted from WM according to behavioural (task) demands (Jensen and Mazaheri 2010; Lundqvist et al. 2023)."

The authors tend to, in particular in the abstract, imply a causal role between alpha and beta bursts and working memory mechanisms. In my opinion, this is unwarranted and should be toned down. What has been found is a relationship between working memory and the measured neural responses. This does not mean that alpha and beta bursts actually regulate working memory processes.

We have rewritten this part of the Abstract as we indeed measure correlates. We now use correlates or that they "reflect the inhibition" rather than actually implementing it. We also changed texts in Introduction and Discussion except on a few occasions where it is clear we are talking about a model/interpretations.

New Abstract:

"Multiple neural mechanisms underlying gating to and from working memory have been proposed, with divergent results obtained in human and animal studies. Previous results from non-human primates suggest prefrontal beta frequency bursts as correlates of transient inhibition during selective encoding. Human studies instead suggest a similar role for sensory alpha power fluctuations. Discrepancies between studies, whether due to differences in analysis, species, or cortical regions, remain unexplained. We addressed this by performing analogous single-trial burst analysis we earlier deployed on non-human primates on human whole-brain electrophysiological activity. Participants performed a sequential working memory task with distractors. Our results reconcile earlier findings by demonstrating that both alpha and beta bursts correlate with the filtering and control of memory items, but with region and task-specific differences between the two rhythms. Occipital beta burst patterns were selectively modulated during the transition from sensory processing to memory retention whereas prefrontal and parietal beta bursts tracked sequence order and were proactively upregulated prior to upcoming target encoding. Occipital alpha bursts instead increased during the actual presentation of unwanted sensory stimuli. Source reconstruction in addition suggested the involvement of striatum and thalamus. Thus specific whole-brain burst patterns correlate with different aspects working memory control. "

The authors report increased beta power bursts before target versus distractor trials and

interpret this as evidence for “down-regulation of existing WM-related activity”. However, WM-specificity was not tested here since the distractors were always presented after a target stimulus. A comparison to distractor trials without a preceding target stimulus would be necessary to make such claims. The effect reported here could be related to preparatory attention (getting ready to encode a target, versus ignoring a distractor).

We had a paragraph in the existing Discussion where we alluded to this. However, we agree that we were not explicit enough. Thus this section is now amended to make clear that one has to decouple regulation of WM from processing of distractors by experimental design.

“We draw these conclusions primarily based on timing of the alpha and beta bursting. Further experiments, where the order of distractors and target items are varied, and in which distractors are not always predictable, may shed further light on this potential distinction between alpha and beta bursting. If, for instance, distractors can appear both before or after relevant information should be encoded, the proposed role of beta in regulating already encoded information may be directly tested.”

The authors use, what they call “bursts” as their main measurement. This needs to be better explained. Why exactly is this method used over the more conventional total power? And how does it differ? I fail to see any advantage or even notable differences for this metric. To be clear, I am not saying that there aren’t any, but if there are, the authors need to better highlight and explain those. Would it be possible to also do the analyses on total power (Figures 4, 5, 6)? Then the results can be more easily compared with the wider literature, while highlighting the potential differences between bursts and power.

We have, remade those indicated figures for power (Figure S1; which is directly comparable to Figure 5, so the main differences and similarities may be inferred). We agree that this helps increase the connection of our results to the literature. See also response to reviewer 2, remark 4. If/Because power is driven by the bursts, the two metrics are highly correlated. At the same time we wished to directly compare data from humans to our previous analysis of bursts in non-human primates.

Yet, if bursts of power underlie changes in power the former metric has several advantages, of whom we take advantage of some but not all here. First of all, it is conceptually important and we believe the insight that oscillations are transient will shape future models of cognition and

change the results interpretation. It suggests more discrete neural processes rather than slowly changing dynamics, this should be reflected in how we think about cognition (we believe it isn't to a large degree in the current literature). Second it provides better signal to noise ratio, as the metric focus on time periods where there is actual periodic activity in a given frequency band (alleviating the contribution of 1/f noise and bleeding from other frequencies). This is evidenced by the fact that power has a similar overall dynamic, but that differences between conditions are often not significant (Figure S1). Third, the extraction of bursts allows us to utilize their timing to relate to behavior or other metrics (here optical tagging, and also now the new analysis of timing between regions) on a moment-by-moment basis. Here, burst and power analysis becomes different even on a single trial case since the former takes into account the timing, so we can observe that power in tagged frequencies is suppressed briefly following a burst. In the extension, it also enables us to look at how beta/alpha activity spreads within cortex to infer directionality of processing (Zich et al., 2023). This can of course be done by time-dependent power estimates, but should be enhanced by metrics that explicitly utilize the discrete nature of power changes (Lundqvist et al., 2024; Hindriks and Tewarie, 2023). Apart from the new figures relevant to this concern (Figure S1; Figure S4) we also briefly motivate the use of burst analysis in Introduction:

"Establishing single-trial correlates of cognition is conceptually important as it has suggested that cognition is supported by brief events rather than slowly changing dynamics. It may also allow us to better capture moment-by-moment interactions between regions as cognitive commands unfold (Hindriks and Tewarie 2023; Zich et al. 2023; Lundqvist et al. 2024)."

Hindriks, R., & Tewarie, P. K. (2023). Dissociation between phase and power correlation networks in the human brain is driven by co-occurrent bursts. *Communications biology*, 6(1), 286.

Lundqvist, M., Miller, E. K., Nordmark, J., Liljefors, J., & Herman, P. (2024). Beta: bursts of cognition. *Trends in Cognitive Sciences*.

Zich, C., Quinn, A. J., Bonaiuto, J. J., O'Neill, G., Mardell, L. C., Ward, N. S., & Bestmann, S. (2023). Spatiotemporal organisation of human sensorimotor beta burst activity. *elife*, 12, e80160.

Minor comments:

In lines 398-399 it is stated "For analysis for which a baseline was applied..." However, as far

as I could tell, it is not mentioned again which analyses used a baseline.

Thank you for pointing this out. This referred to the calculation of z-scores which is presented in several places, we have clarified this.

Figure 1: The last delay, after the target 4 is 0.5 s in duration. The figure caption and the methods state that it is 750ms, however.

Thank you, this was incorrect in the Figure and correct on the other places. We have corrected this.

Reviewer #3 (Remarks on code availability):

The scripts to reproduce figures seem to be all there, as well as a readme file with instructions. However, no data has been uploaded, so the scripts cannot be run.

The data (down sampled, epoched MEG data), and updated code has now been uploaded. Our initial thought was to upload the data upon publication.

REVIEWERS' COMMENTS

Reviewer #1 (Remarks to the Author):

I thank the authors for addressing all my previous points, and I agree that the manuscript is now significantly improved. I value their careful consideration of my concerns and the inclusion of new results. However, I still believe that the single trial burst framework is somewhat underutilized, as even more strongly highlighted now by the power results in Figure S1 (which is an important add-on). Most, if not all, of these analyses could have been conducted using single-trial power analyses. The results in Figure S4 are promising, as these short burst events are well-suited to investigate the directionality of information flow, which is challenging to do with single-trial power. However, the results are presented descriptively without providing any statistics or correlation to behavior. Therefore, I do not suggest including it as a main figure. Nonetheless, since the authors address a timely question and their results raise important conceptual questions, I believe the manuscript will be of significant interest to the field.

Minor points:

- Please provide beta-values instead of t-values for all GLM results, as beta-values are more commonly reported.
- Cluster-based permutation results: In line 194, p-values smaller than $1e-6$ are mentioned. How is this possible with only 10,000 permutations, where the smallest possible p-value would be $1e-4$? This issue also affects results mentioned later.
- Relation of burst rate to behavior (line 290 and following): Please specify in the text which region and which time windows were used for this analysis. This was not entirely clear from the text.
- The manuscript contains many typos and incorrect sentences throughout. I suggest a thorough proofreading.

Reviewer #1 (Remarks on code availability):

The scripts are well documented and structured. A ReadMe detailing how to run the analyses was provided and is written clearly.

Reviewer #2 (Remarks to the Author):

Thanks to the authors for their thoughtful and comprehensive reply. I have no further concerns.

REVIEWERS' COMMENTS

We would once again like to thank the reviewers for their time and feedback on the revised manuscript. We greatly appreciate the valuable insights they have provided. Below, we provide a response to each of the reviewers' concerns.

Reviewer #1 (Remarks to the Author):

I thank the authors for addressing all my previous points, and I agree that the manuscript is now significantly improved. I value their careful consideration of my concerns and the inclusion of new results. However, I still believe that the single trial burst framework is somewhat underutilized, as even more strongly highlighted now by the power results in Figure S1 (which is an important add-on). Most, if not all, of these analyses could have been conducted using single-trial power analyses. The results in Figure S4 are promising, as these short burst events are well-suited to investigate the directionality of information flow, which is challenging to do with single-trial power. However, the results are presented descriptively without providing any statistics or correlation to behavior. Therefore, I do not suggest including it as a main figure. Nonetheless, since the authors address a timely question and their results raise important conceptual questions, I believe the manuscript will be of significant interest to the field.

We agree that figure S4 could be further enhanced, and have added statistical measures (non-parametric permutation tests) strengthening and confirming the findings.

Minor points:

- Please provide beta-values instead of t-values for all GLM results, as beta-values are more commonly reported.

We now provide values for the coefficient in addition to the other statistical results.

- Cluster-based permutation results: In line 194, p-values smaller than $1e-6$ are mentioned. How is this possible with only 10,000 permutations, where the smallest possible p-value would be $1e-4$? This issue also affects results mentioned later.

Thank you for identifying this mistake. This should have shown $p < 0.001$ and has been corrected.

- Relation of burst rate to behavior (line 290 and following): Please specify in the text which region and which time windows were used for this analysis. This was not entirely clear from the text.

We now write:

"The model had subject as random effect, condition, serial order of probed item, and occipital burst count during encoding (onset – 500ms) as fixed effects."

- The manuscript contains many typos and incorrect sentences throughout. I suggest a thorough proofreading.

We did indeed find typos and two incomplete sentences.

Reviewer #1 (Remarks on code availability):

The scripts are well documented and structured. A ReadMe detailing how to run the analyses was provided and is written clearly.

Thank you.

Reviewer #2 (Remarks to the Author):

Thanks to the authors for their thoughtful and comprehensive reply. I have no further concerns.

We thank you for your time, effort and valuable input.